# Cell-specific bioorthogonal tagging of glycoproteins

Anna Cioce [1,2], Beatriz Calle [1,2,3], Tatiana Rizou[3,18], Sarah C. Lowery[4,18], Victoria L. Bridgeman[3,18], Keira E. Mahoney [4,18], Andrea Marchesi[1,2], Ganka Bineva-Todd[2], Helen Flynn [5], Zhen Li[1,2], Omur Y. Tastan[2], Chloe Roustan[6], Pablo Soro-Barrio [7], Mahmoud-Reza Rafiee[8], Acely Garza-Garcia [9], Aristotelis Antonopoulos [10], Thomas M. Wood[11,14], Tessa Keenan [12], Peter Both[13,15], Kun Huang[13,16], Fabio Parmeggian [13,17], Ambrosius P. Snijders [5], Mark Skehel [5], Svend Kjær [6], Martin A. Fascione [12], Carolyn R. Bertozzi [11], Stuart M. Haslam [10], Sabine L. Flitsch [13], Stacy A. Malaker [4], Ilaria Malanchi [3] & Benjamin Schumann [1,2] ✉

Altered glycoprotein expression is an undisputed corollary of cancer development. Understanding these alterations is paramount but hampered by limitations underlying cellular model systems. For instance, the intricate interactions between tumour and host cannot be adequately recapitulated in monoculture of tumour-derived cell lines. More complex co-culture models usually rely on sorting procedures for proteome analyses and rarely capture the details of protein glycosylation. Here, we report a strategy termed Bio-Orthogonal Cell line-specific Tagging of Glycoproteins (BOCTAG). Cells are equipped by transfection with an artificial biosynthetic pathway that transforms bioorthogonally tagged sugars into the corresponding nucleotide-sugars. Only transfected cells incorporate bioorthogonal tags into glycoproteins in the presence of non-transfected cells. We employ BOCTAG as an imaging technique and to annotate cell-specific glycosylation sites in mass spectrometry-glycoproteomics. We demonstrate application in co-culture and mouse models, allowing for profiling of the glycoproteome as an important modulator of cellular function.

Cancer is a multifactorial disease consisting of an interplay between host and tumour. Emulating the complexity of a tumour in cell monoculture is thus incomplete by design, requiring more elaborated co-culture systems or in vivo models[1–3]. Recent years have seen a stark increase in methods to probe the transcriptomes of tumour and host cell populations, providing some insight into their state within a multicellular conglomerate[4]. However, the relationship between transcriptome and proteome is still elusive[5]. In addition, posttranslational modifications (PTMs) heavily impact the plasticity of the proteome. Glycosylation is the most complex and most abundant PTM, but challenging to probe due to the non-templated nature of glycan

biosynthesis[6]. Glycans are generated by the combinatorial interplay of >250 glycosyltransferases (GTs) and glycosidases, mostly in the secretory pathway[7]. A small number of glycoproteins aberrantly expressed in cancer, such as mucins, have been approved as diagnostic markers, but their discovery is a particular challenge[8,9]. This is especially true when in vivo or in vitro model systems are comprised of cell populations from the same organism that do not allow distinction of proteomes by amino acid sequence[10,11]. Methods to study the glycoproteome of a cell type in co-culture or in vivo are therefore an unmet need.

Metabolic oligosaccharide engineering (MOE) produces chemical reporters of glycan subtypes[12]. MOE reagents are membrane-permeable

monosaccharide precursors modified with chemical tags amenable to bioorthogonal chemistry[13]. Following incorporation into the glycoproteome, chemical tags are reacted with traceable enrichment handles or fluorophores, for instance by Cu(I)-catalysed azide-alkyne cycloaddition (CuAAC)[14,15]. Many MOE reagents are based on analogues of sugars such as N-acetylgalactosamine (GalNAc) that are straightforward to chemically tag by replacing the acetamide with bioorthogonal N-acylamides (Fig. 1a). Unmodified GalNAc is normally activated by the biosynthetic GalNAc salvage pathway to the nucleotide-sugar UDP-GalNAc that can follow two major distinct metabolic fates (Fig. 1a)[14,16–18]. First, the 20 members of the GalNAc transferase family (GalNAc-T1…T20) use UDP-GalNAc to form the linkage GalNAcα-Ser/Thr and thereby prime cancer-relevant O-GalNAc glycans[14,19,20]. Second, epimerisation at the GalNAc C4 position by the UDP-galactose-4-epimerase (GALE) yields UDP-N-acetylglucosamine (UDP-GlcNAc) that can be incorporated into different glycan subtypes, for instance Asn-linked N-glycans[17,18,21]. Certain chemical modifications at the N-acyl moiety can render GalNAc analogues recalcitrant to these metabolic processes. For instance, analogues of UDP-GalNAc with long alkyne-containing N-acyl substituents are not biosynthesised by the GalNAc salvage pathway and not used as substrates by Wild Type (WT)-GalNAc-Ts[18,22–24]. While being a substantial impediment to generating MOE reporters, we realised that overcoming these metabolic roadblocks might enable programmable bioorthogonal glycoprotein tagging. Such a strategy would allow for studying the glycoproteome in a cell-specific fashion, which is currently elusive despite the rapid advances in the development of new MOE reagents.

Here, we develop a technique called Bio-Orthogonal Cell-specific Tagging of Glycoproteins (BOCTAG). The strategy uses an artificial biosynthetic pathway to generate alkyne-tagged UDP-GalNAc and UDP-GlcNAc analogues from a readily available GalNAc precursor that is not accepted by the GalNAc salvage pathway. We find that a single methylene group between 5-carbon (GalNAlk) and 6-carbon (GalN6yne) N-acyl substituents drastically reduces uptake by the native GalNAc salvage pathway and thereby reduces the background of bioorthogonal labelling in non-transfected cells. Only cells carrying the artificial pathway biosynthesise the corresponding UDP-sugars (UDP-GalN6yne and UDP-GlcN6yne) that are then used by GTs to chemically tag the glycoproteome. We further expand the strategy with mutant GalNAc-Ts that are engineered to accept UDP-GalN6yne as a substrate. The combined use of an artificial biosynthetic pathway and engineered GalNAc-Ts enables GalN6yne-mediated fluorescent labelling of the cellular glycoproteome that is two orders of magnitude higher than in cells carrying neither component. We demonstrate that BOCTAG allows for programmable glycoprotein tagging in co-culture and mouse models. Moreover, the nature of the artificial biosynthetic pathway allowed for the use of readily available Ac4GalN6yne as a precursor with enhanced stability over previously used caged GalN6yne-1-phosphates as an essential pre-requisite for in vivo applications. We show that the chemical modification enters a range of glycan subtypes, supporting the use of BOCTAG to tag a large number of glycoproteins in complex biological systems.

## Results

### Developing an artificial biosynthetic pathway for chemically tagged UDP-sugars

The human GalNAc salvage pathway consists of the kinase GALK2 and the pyrophosphorylases AGX1/2 to convert GalNAc first into GalNAc-1-phosphate and subsequently into UDP-GalNAc, respectively (Fig. 1a). Since analogues of GalNAc nor GalNAc-1-phosphate can be utilised by any other metabolic enzyme, the GalNAc salvage pathway was deemed suitable for monitoring conversions of each step while supplying readily accessible synthetic, bioorthogonal precursors. GALK2 and AGX1/2 cannot accomodate chemical modifications at the N-acyl moiety of GalNAc (Fig. 1a), corroborated by crystal structures of these enzymes (Supplementary Fig. 1)[18,24–26]. An artificial biosynthetic pathway was thus designed to convert chemically tagged GalNAc analogues first to the corresponding sugar-1-phosphates and subsequently to the UDP-sugars. We chose both a 6-carbon hex-5-ynoate chain (GalN6yne) and a 5-carbon pent-4-ynoate chain (GalNAlk) as GalNAc modifications due to their availability and previous use by us and others[18,24,27]. According to in vitro enzymatic assays detected by Liquid Chromatography-Mass Spectrometry (LC-MS), recombinant GALK2 accepted GalNAlk as a substrate, but only marginally accepted GalN6yne (Fig. 1b). In contrast, promiscuous bacterial N-acetylhexosaminyl kinases (NahK) from various source organisms converted GalN6yne to GalN6yne-1-phosphate almost quantitatively (Fig. 1b, Supplementary Fig. 2a)[28]. Similarly, the WT pyrophosphorylase AGX1 showed little to no turnover of both GalNAlk-1-phosphate and GalN6yne-1-phosphate (Supplementary Methods) to the corresponding UDP-sugars (Fig. 1b). We and others have mutated AGX1 at residue Phe383 to smaller amino acids to accommodate chemical N-acyl modifications[24,29]. AGX1[F383A], herein called mut-AGX1, converted both synthetic GalNAlk-1-phosphate and GalN6yne-1-phosphate to UDP-GalNAlk and UDP-GalN6yne, respectively (Fig. 1b).

We next assessed UDP-sugar biosynthesis in the living cells. Stable bicistronic expression of a codon-optimised version of *Bifidobacterium longum* NahK as well as mut-AGX1 in K-562 cells biosynthesised UDP-GalNAlk and UDP-GalN6yne from membrane-permeable per-acetylated precursors Ac4GalNAlk and Ac4GalN6yne, respectively (Fig. 1c). Expression of either enzyme alone or WT-AGX1 leds to inefficient biosynthesis compared to levels of native UDP-sugars (Supplementary Fig. 3). We confirmed these results by feeding cells a caged precursor of GalN6yne-1-phosphate that was uncaged in the living cells and converted to UDP-GalN6yne only in the presence of mut-AGX1 (Supplementary Fig. 3). Alkyne-tagged UDP-GalNAc analogues were converted to the corresponding UDP-GlcNAc analogues (UDP-GlcNAlk or UDP-GlcN6yne, respectively) in cells by the epimerase GALE, which was further corroborated in an in vitro epimerisation assay (Fig. 1a, c and Supplementary Figs. 2b, 3). Thus, installing an artificial biosynthetic pathway led to programmable biosynthesis of alkyne-tagged analogues of UDP-GalNAc and UDP-GlcNAc.

We next assessed chemical tagging of the cell surface glycoproteome in living cells. K-562 cells stably expressing combinations of NahK and AGX1 were fed with DMSO, Ac4GalNAlk or Ac4GalN6yne and reacted with the clickable fluorophore CF680-picolyl azide by CuAAC. The MOE reagent Ac4ManNAlk (tetra-acetylated N-(4-pentynoyl)-mannosamine) that enters the pool of the sugar sialic acid was included as a positive control. Alkyne tags were visualised by in-gel fluorescence after cell lysis (Fig. 2a). While Ac4GalNAlk feeding led to high-intensity fluorescent signal when NahK and mut-AGX1 were expressed, substantial signal was observed in cells expressing WT-AGX1 when NahK was present (Fig. 2a). Fluorescent signal after Ac4GalNAlk feeding was also observed in cells transfected with an empty plasmid or only overexpressing WT-AGX1, confirming the permissiveness of the GalNAc salvage pathway for GalNAlk (Fig. 1b)[18]. In contrast, Ac4GalN6yne incorporation was critically dependent on the expression of mut-AGX1, while the presence of NahK led to a further sixfold increase in fluorescence intensity (Fig. 2a). Ac4ManNAlk-fed cells produced fluorescent signal regardless of the enzyme combination expressed. Dose-response experiments showed that Ac4GalN6yne-mediated fluorescence intensity increased over two orders of magnitude with the concentration of the probe between 16 nM and 50 μM only when NahK and mut-AGX1 were present (Fig. 2b).

### Effect of an artificial biosynthetic pathway for UDP-sugars on transcriptome and glycome

Transfection and feeding with chemically modified sugars can, in theory, alter the cellular transcriptome, leading to artefacts in protein expression and metabolic labelling. We performed transcriptomic

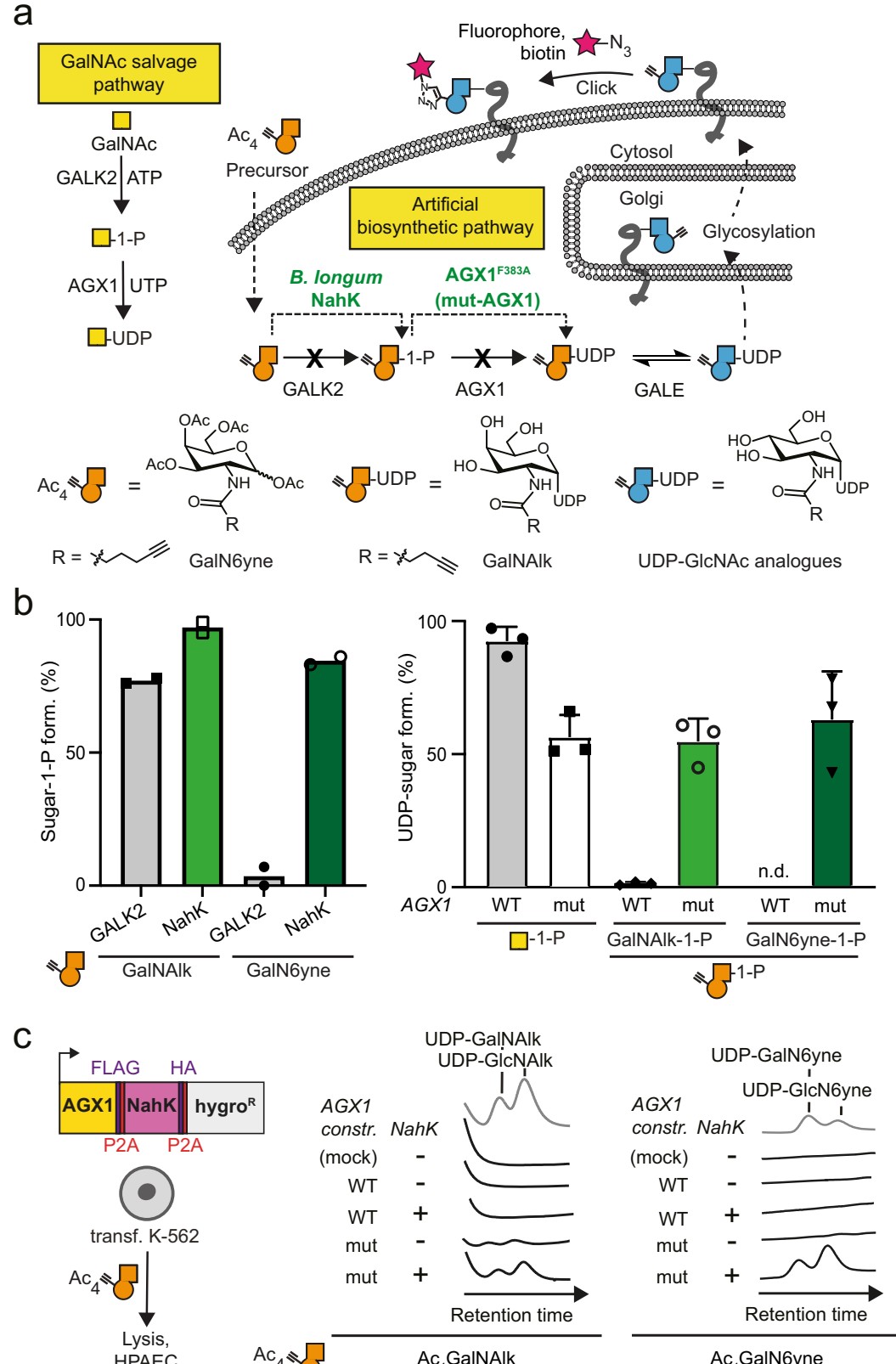

analyses in cells transfected with either NahK/mut-AGX1 or empty plasmid, and fed with either DMSO vehicle, Ac₄GalN6yne or Ac₄Gal-NAc. Correlation plot and principal component analysis (PCA, Supplementary Fig. 4) showed that the day of sample collection has a greater effect on transcript levels than either transgene expression or compound treatment (Supplementary Fig. 4b). These data suggest

that neither artificial biosynthetic pathway nor compound feeding has substantial effects on the transcriptome. We further measured the levels of endogenous UDP-sugars and found no substantial changes upon expression of mut-AGX1, regardless of feeding with DMSO vehicle or Ac₄GalN6yne. Similarly, co-expression of NahK and mut-AGX1 led to no changes in native UDP-sugar levels when fed with

**Fig. 1 | Development of an artificial biosynthetic pathway for chemically tagged UDP-GalNAc/GlcNAc analogues. a** Strategy of metabolic oligosaccharide engineering. A chemically modified GalNAc analogue that is not accepted by the GalNAc salvage pathway should be processed by an artificial biosynthetic pathway. *B. longum* NahK and mut-AGX1 biosynthesise UDP-GalNAc analogues and, by epimerisation, UDP-GlcNAc analogues. Incorporation into glycoconjugates can be traced by CuAAC. **b** In vitro evaluation of GalNAc-1-phosphate analogue synthesis by human GALK2 or *B. longum* NahK (left) and UDP-GalNAc analogue synthesis by WT- or mut-AGX1 (right). Data were recorded in Liquid Chromatography-Mass Spectrometry (LC-MS) assays and processed by integrated ion counts against a calibration curve or a single run of authentic product. Data are from two independent experiments and depicted as individual data points and means (left) or from three independent experiments and depicted as means + standard deviation (SD, right) overlaid to the individual data points. **c** Biosynthesis of UDP-GalNAc/GlcNAc analogues in cells stably expressing both NahK and mut-AGX1 or either component, as assessed by High Performance Anion Exchange Chromatography (HPAEC). A hygromycin resistance gene allows for stable transfection. Data are representative of one out of two independent experiments collected on two different days. Mock: pSBbi-GH empty plasmid. Source data are provided as a Source Data file.

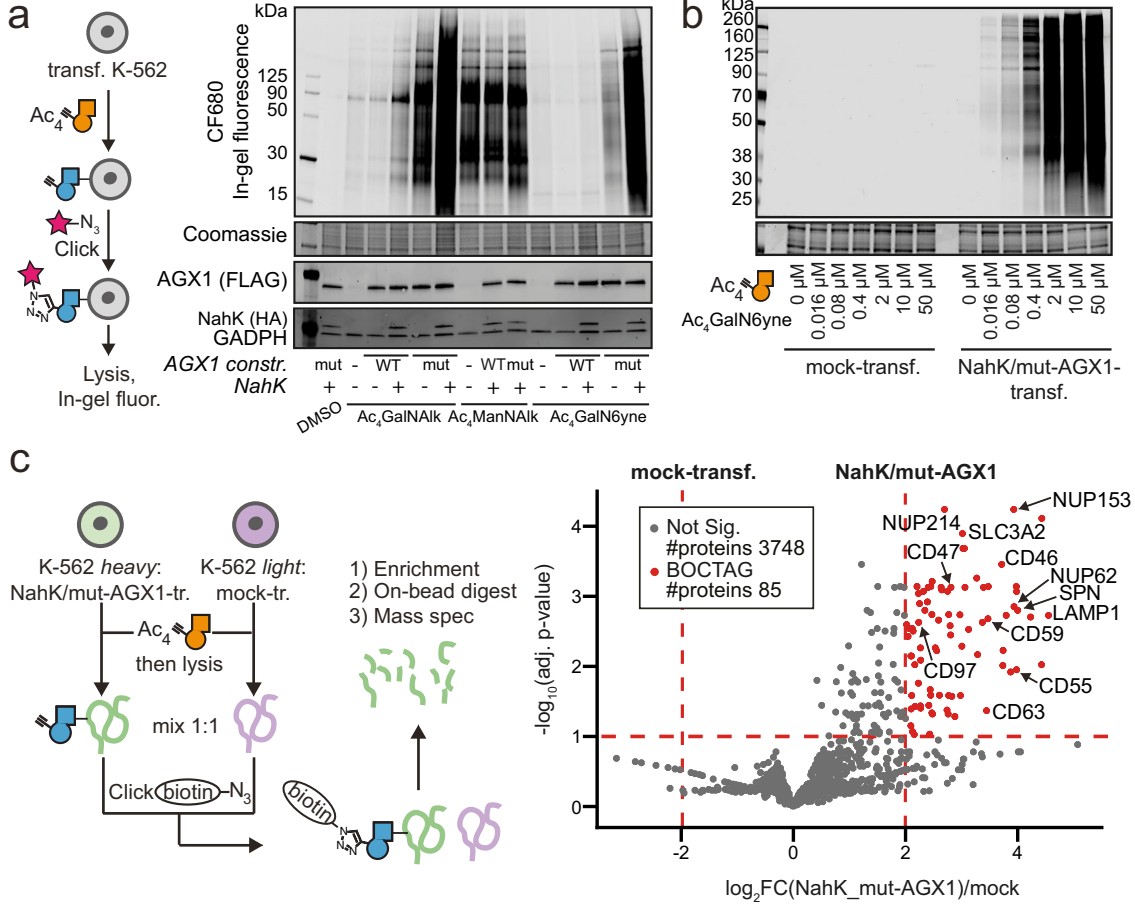

**Fig. 2 | An artificial biosynthetic pathway enables programmable chemical tagging of the glycoproteome. a** Evaluation of cell surface glycoproteome tagging after treating K-562 cells stably expressing NahK/AGX1 combinations with 50 μM Ac₄GalNAlk, 50 μM Ac₄GalN6yne or 10 μM Ac₄ManNAlk. Glycoproteins were visualised by in-gel fluorescence after treating cells with CF680-picolyl azide under CuAAC conditions and subsequent cell lysis. **b** Dose-response experiment of cell surface glycoproteome tagging, with samples processed as in (**a**). Data in (**a** and **b**) are representative of one out of two independent experiments. **c** Quantitative measurement of glycoprotein tagging by Stable Isotope Labelling by Amino Acids in Cell Culture (SILAC)-based proteomics. Data were analysed from three independent experiments, collected on three different days, with forward (heavy mock, light NahK/mut-AGX1) and reverse (light mock, heavy NahK/mut-AGX1) analyses incorporated as a total of six replicates. Differentially enriched proteins were determined using empirical Bayes moderated *t*-test (two-sided) by limma package. Multiple-testing corrections were performed using the Benjamini-Hochberg procedure to calculate adjusted *p* values (False Detection Rate). Data are visualised as volcano plot, choosing fourfold enrichment and a *p* value of 0.1 as cutoffs, with example glycoproteins annotated. Significance levels were indicated. Mock: pSBbi-GH empty plasmid. Source data are provided as a Source Data file.

Ac₄GalN6yne (Supplementary Fig. 5a). Levels of UDP-GlcNAc/GalNAc were increased by 33/36%, 28/30%, and 26/30% in WT-AGX1 samples (DMSO- and Ac₄GalN6yne-fed) and DMSO-fed NahK/mut-AGX1, respectively. The relative ratio of both metabolites was constant in all cell lines (Supplementary Fig. 5b). We also found no differences in concentrations of the metabolite cytidine monophosphate-*N*-acetylneuraminic acid (CMP-Neu5Ac). To assess if the cellular glycome would be affected by variations in nucleotide-sugar concentrations, we performed lectin blotting on lysates of cells expressing NahK/mut-AGX1. Compared to mock-transfected cells, we found no differences in binding patterns with the four lectins ConA, MAL II, SNA and AAL, as well as the antibody RL2 which detects nucleocytoplasmic O-linked GlcNAc (Supplementary Fig. 6). We also established that chemical tagging is measurable in a dose-dependent fashion both on the cell surface (when CuAAC was performed prior to lysis) and in lysate (Supplementary Fig. 7). Due to the robustness of metabolic incorporation, we used Ac₄GalN6yne as the MOE reagent for all subsequent applications of BOCTAG.

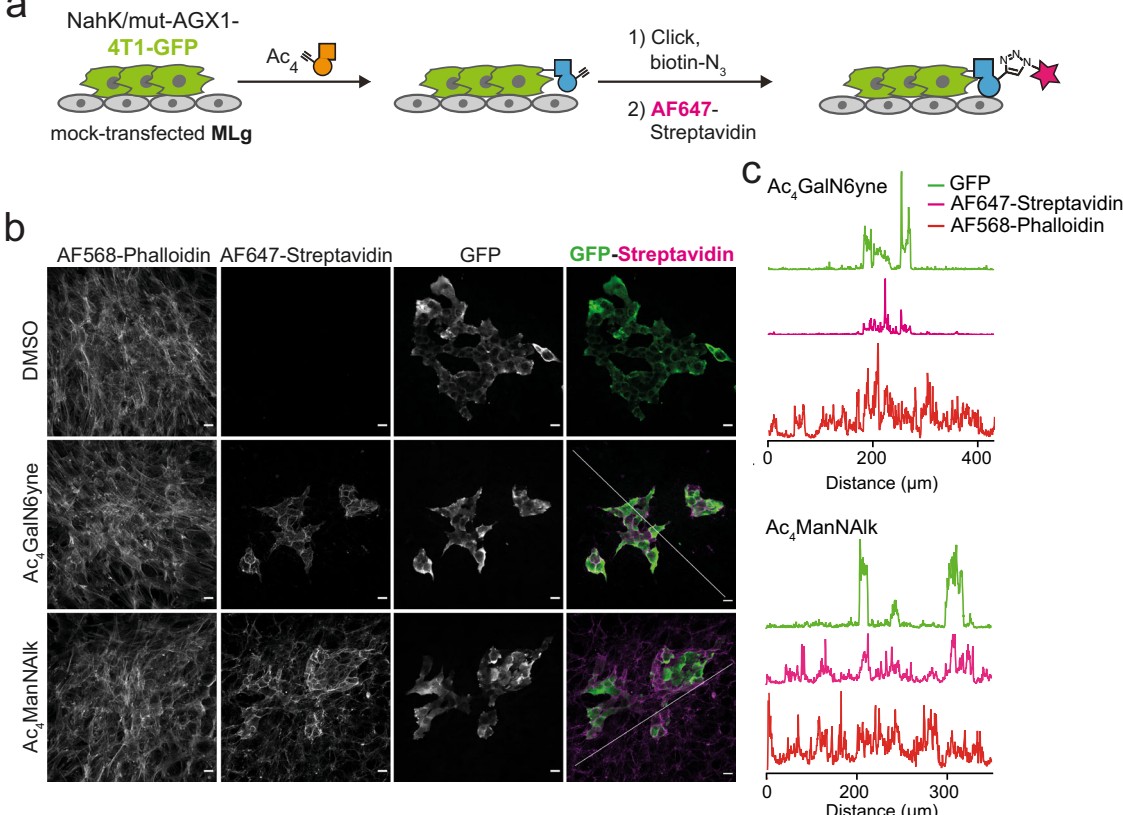

**Fig. 3 | Bioorthogonal cell-specific glycoprotein tagging in co-culture.**
**a** Schematic of the 4T1-MLg co-culture experiment. Green fluorescent protein (GFP)-expressing 4T1 cells transfected with NahK/mut-AGX1 should be selectively positive for AlexaFluor647-labelling in BOCTAG. **b** Fluorescence microscopy, using co-cultures fed with 50 µM Ac₄GalN6yne or 50 µM Ac₄ManNAlk as well as AlexaFluor568-Phalloidin as a counterstain. Scale bar, 20 µm. **c** Intensity profiles of

fluorescent signal between GFP and AF647 in Ac₄GalN6yne- (top) or Ac₄ManNAlk-fed (bottom) co-cultures. The intensity profiles of GFP, AF647-Streptavidin and AF568-Phalloidin signals were measured along a diagonal line drawn across the fluorescent image. Data are representative of one out of two independent experiments. Source data are provided as a Source Data file.

## An artificial biosynthetic pathway allows for programmable enrichment of the glycoproteome

We next assessed the ability of BOCTAG to selectively tag the (glyco-) proteome of cells transfected with NahK and mut-AGX1. K-562 cells transfected with NahK/mut-AGX1 or an empty plasmid (mock-transfected) were individually grown in heavy or light media in the presence of either Ac₄GalN6yne or DMSO. Lysates of these cells were mixed as different combinations to contain equal amounts of heavy and light protein, and clickable biotin-picolyl azide was installed on tagged glycoproteins by CuAAC. Enrichment on neutravidin beads followed by on-bead digest allowed analysis by quantitative mass spectrometry (MS). In three independent experiments, in which both combinations of heavy and light SILAC labelling each were used (Fig. 2c), we found peptides from 85 proteins to be significantly enriched in NahK/mut-AGX1-transfected cells (Supplementary Data 1). More than 98% (84/85) of these proteins have been previously annotated[30–32] as either N- or O-glycosylated, including the nucleoporins Nup62 and Nup153 as well as the cell surface proteins CD47 and NOTCH1, confirming the effectiveness of the approach for tagging glycoproteins.

## Cell type-specific glycoproteome tagging in co-culture

We next assessed the suitability of the artificial biosynthetic pathway NahK/mut-AGX1 as a BOCTAG cell type-specific glycoproteome labelling technique for visualization by fluorescence microscopy. Colonies of NahK/mut-AGX1-transfected and GFP-expressing 4T1 murine breast cancer cells were established on a monolayer of non-transfected MLg murine fibroblast cells. After co-culturing for 72 h, cells were treated

with either Ac₄GalN6yne, Ac₄ManNAlk or DMSO (Fig. 3a). Clickable biotin-picolyl azide was installed by CuAAC followed by Streptavidin-AF647 staining to visualise chemical tagging, and cells were counter-stained with fluorescently labelled phalloidin. Streptavidin-AF647 signal was strongly and reproducibly restricted to GFP-expressing cells only when Ac₄GalN6yne was fed and NahK/mut-AGX1 were expressed (Fig. 3b, c and Supplementary Figs. 8–11), indicating a localised BOCTAG signal. In contrast, the promiscuous MOE reagent Ac₄ManNAlk was non-specifically incorporated throughout the entire co-culture (Fig. 3b, c, Supplementary Figs. 8–11). When both GFP-4T1 and MLg cell lines expressed NahK/mut-AGX1 and were fed with Ac₄GalN6yne, both exhibited a strong Streptavidin-AF647 signal (Supplementary Fig. 11b). Taken together, BOCTAG enables cell-specific tagging of cell surface glycoproteins in co-culture.

## Assessing and manipulating the glycan types tagged by GalN6yne

We next sought to assess and expand the glycan subtypes targeted by our MOE approach. We were prompted by our recent findings that GalNAc analogues with bulky *N*-acyl chains such as GalN6yne are not incorporated into O-GalNAc glycans by WT-GalNAc-Ts (Fig. 4a)[23,24,33]. We have created GalNAc-T mutants termed BH-GalNAc-Ts (for "Bump-and-Hole engineering", the process used to design the mutants) that selectively use chemically tagged UDP-GalNAc analogues in glycosylation reactions[23,24,33]. We stably co-expressed WT- or BH-versions of GalNAc-T1 or T2 from plasmids also encoding NahK and mut-AGX1 in K-562 cells (Fig. 4a). Expression of BH-GalNAc-Ts increased the

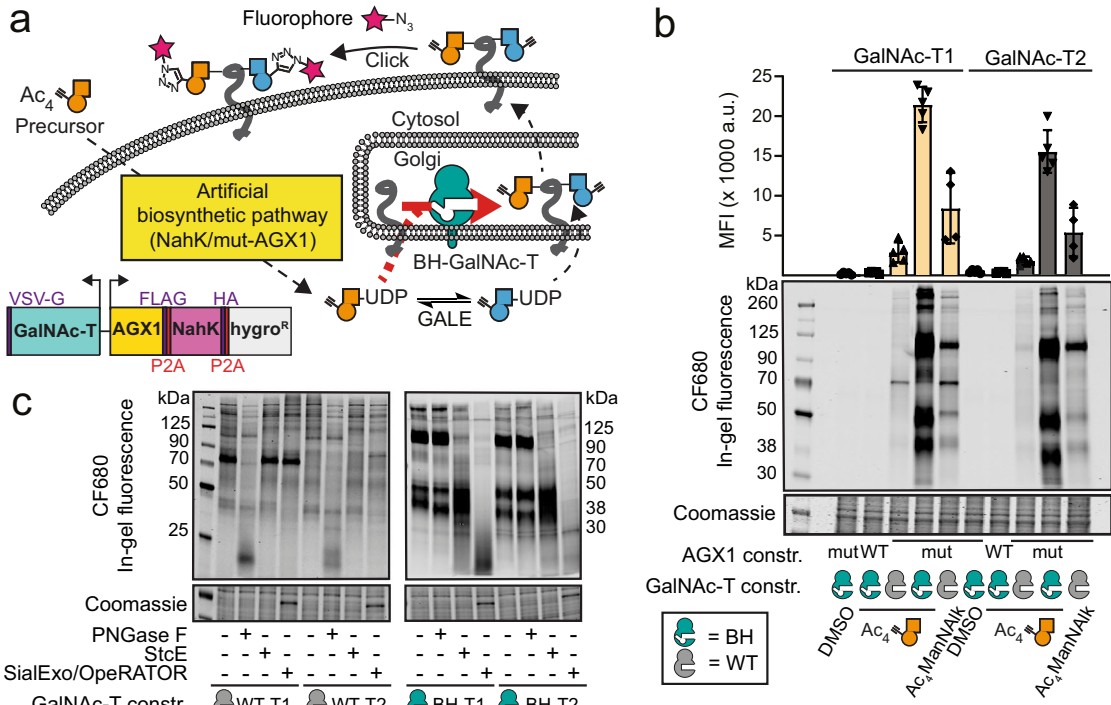

**Fig. 4 | Enhancement of programmable glycoprotein tagging by expression of BH-GalNAc-Ts. a** Strategy of expanding glycoprotein tagging to include O-GalNAc glycans. Expression of BH-GalNAc-Ts selectively engineered to accommodate bulky chemical tags enhances O-GalNAc tagging in cells expressing NahK/mut-AGX1. **b** Evaluation of tagging efficiency by feeding transfected K-562 cells with either DMSO, 1 μM Ac₄GalN6yne or 2 μM Ac₄ManNAlk. Tagging was analysed by in-gel fluorescence and quantification by densitometry. Data are depicted as individual points and means ± SD from at least four independent experiments. **c** Assessment of tagged glycan subtypes by treating the samples of cells fed with Ac₄GalN6yne analysed in (**b**) with hydrolytic enzymes. Gel regions with lysates of WT- and BH-GalNAc-T1/T2 transfected cells are scanned independently. Gel images are shown with different intensities to best visualise the effect of enzyme treatment. Data are representative of one out of a total of four replicate labelling experiments performed on two different days. Source data are provided as a Source Data file.

intensity of in-gel fluorescence more than sevenfold over WT-GalNAc-Ts when cells were fed with Ac₄GalN6yne (Fig. 4b). WT-AGX1 expressing cells lacked UDP-GalN6yne/UDP-GlcN6yne biosynthesis (Fig. 1c) and did not show any discernible fluorescent signal over vehicle control DMSO. We assessed the subtypes of the chemically tagged glycans by digestion with the hydrolytic enzymes PNGase F (removes N-glycosylation), StcE (digests mucin-type glycoproteins) and OpeRATOR (digests O-GalNAc glycoproteins in the presence of the sialidase SialEXO) prior to in-gel fluorescence[34]. In cells expressing NahK, mut-AGX1 and WT-GalNAc-Ts, fluorescent labelling was sensitive to PNGase F treatment, indicating that the major target structures are N-glycoproteins in these cells (Fig. 4c). Incomplete signal abrogation by PNGase F could result from incorporation of GlcN6yne in the protein-proximal core of N-linked glycans, which would be functionalised with fluorophore before digest. Co-expression of BH-GalNAc-Ts led to additional highly intense fluorescent signal of a small number of O-glycoproteins with sensitivity to both StcE and OpeRATOR/SialEXO (Fig. 4c). Thus, BH-GalNAc-Ts broaden the target scope of chemical tagging to include O-GalNAc glycoproteins with high incorporation efficiency. In accordance with this finding, we performed quantitative MS-proteome analysis by SILAC of cell lines expressing NahK/mut-AGX1/BH-GalNAc-T2 (BH-T2). In contrast to cells expressing NahK/mut-AGX1/WT-T2 (Fig. 2c), we observed an increase from 50% to 61% of previously annotated O-GalNAc glycoproteins in the enriched protein fraction (Supplementary Data 1)[30–32]. Concomitant with GalN6yne incorporation into the O-GalNAc glycoproteome, we found a relative reduction of certain glycans containing 2,3-linked sialic acid upon BH-GalNAc-T2 expression in lectin blot analyses (Supplementary Figs. 12, 13) and by MS (Supplementary Figs. 14, 15). Using a doxycycline-inducible expression system[24], we confirmed this tendency with WT-GalNAc-T2 in a titratable fashion (Supplementary

Fig. 13), indicating that overexpression of GalNAc-Ts moderately alters the cellular glycome. Taken together, overexpression of BH-GalNAc-Ts increases incorporation of chemically tagged glycans by BOCTAG by an order of magnitude, allowing entry into the O-GalNAc glycoproteome.

## MS-based validation of cell-type specific labelling in co-culture models

We then validated BOCTAG as a strategy for cell-specific MS-glycoproteome analysis. We chose a co-culture model between murine 4T1 and human MCF7 breast cancer cell lines, opting to distinguish labelled glycoproteins with species-specific peptide sequences by Label-free Quantitative (LFQ) LC/MS-MS analysis. We transfected cells with either a plasmid encoding NahK/mut-AGX1/BH-GalNAc-T2 (termed "BOCTAG-T2") or empty plasmid (pSBbi-Hyg, mock), co-cultured murine and human cells overnight and subsequently fed the co-cultures with either Ac₄GalN6yne or vehicle DMSO. Chemically tagged glycoproteins in the secretome were reacted with acid-cleavable biotin-picolyl azide by CuAAC and enriched on neutravidin magnetic beads (Fig. 5a). On-bead digest yielded a peptide fraction and left glycopeptides bound to beads to be separately eluted with formic acid[24,35,36]. Peptide samples were analysed by LFQ MS-proteomics in two independent experiments, choosing an eightfold enrichment and a p value of 0.1 as cut-offs. We observed species-specific protein enrichment: BOCTAG-T2-expressing 4T1 cells led to 132 selectively enriched murine peptides while BOCTAG-T2-expressing MCF7 cells allowed detection of 24 selectively enriched human peptides when co-cultured with mock-transfected cells of the respective other species (Fig. 5b, Supplementary Data 2). Only two human peptides and one murine peptide were found in the enriched datasets from the corresponding other species. BOCTAG-T2 allows for cell-specific glycosylation site

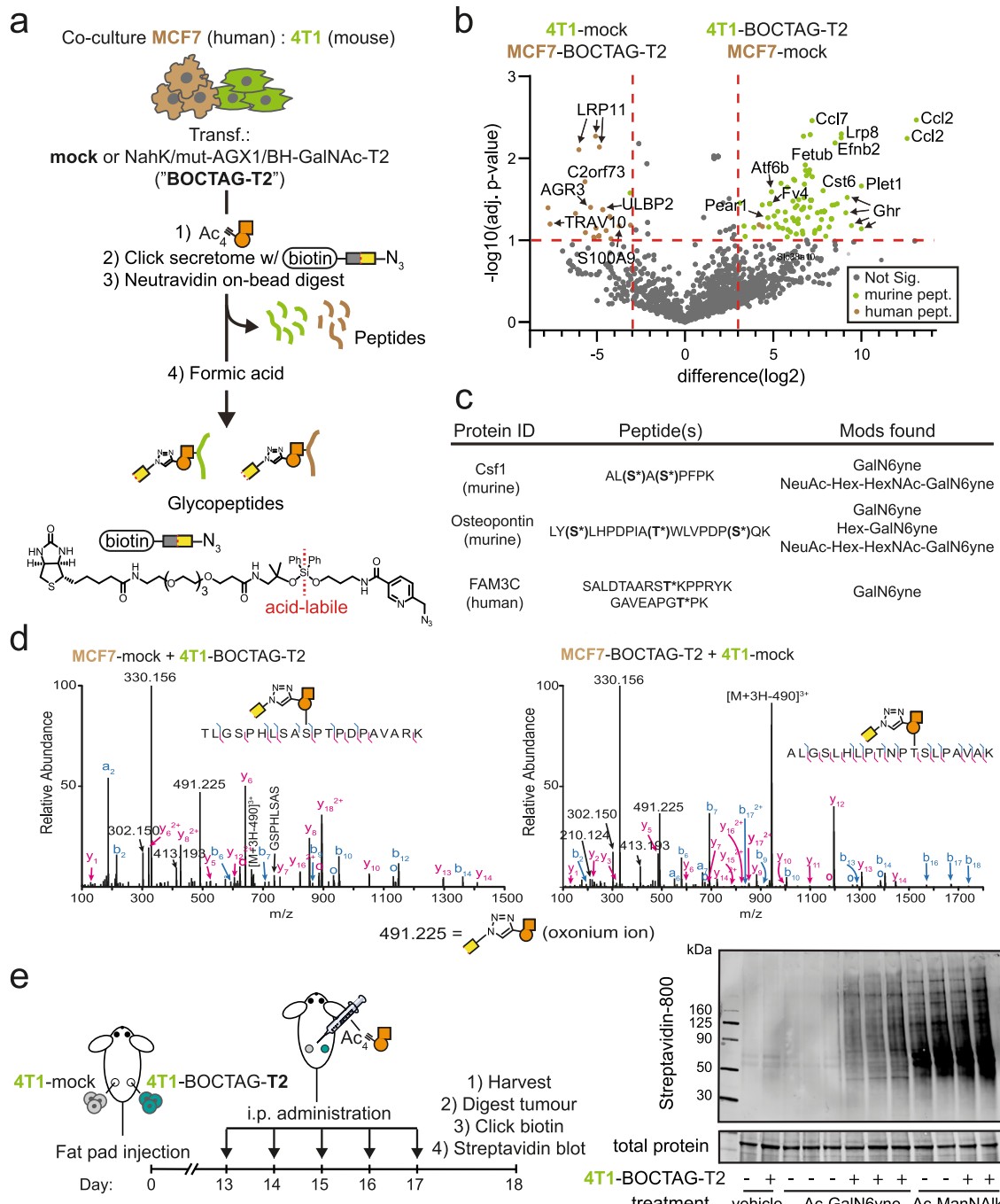

**Fig. 5 | BOCTAG labels glycoproteins in a cell-specific manner in co-culture and in vivo. a** Cell-selective enrichment and MS-glycoproteome analysis of murine-human co-culture systems. MCF7 and 4T1 cells transfected as indicated were co-cultured overnight and treated with DMSO or 10 μM Ac$_4$GalN6yne for 24 h. Secretome was subjected to CuAAC with acid-cleavable biotin-picolyl azide and enriched on neutravidin beads. On-bead digest yielded peptide fractions while acid treatment of beads yielded glycopeptide fractions. **b** MS analysis of peptide fractions from (**a**). Differentially enriched peptides were determined using Welch's *T*-test (two-sided) by using Perseus software. Data are visualised as volcano plot, choosing eightfold enrichment and a *p* value of 0.1 as cut-offs with example peptides (from glycosylated proteins) annotated. Species-specific peptides are indicated. Data are from two independent experiments. **c** Examples of enriched glycopeptides and glycoforms. Asterisks annotate glycosylation sites; parentheses

indicate potential glycosylation sites that could not be confidently assigned. **d** HCD spectra of homologous glycopeptides (Lysosomal Pro-X carboxypeptidase) from murine (left) and human (right) origins. Glycan attachment sites were characterised by ETD (Supplementary Fig. 16). **e** In vivo glycoproteome tagging by BOCTAG-T2. Tumours were grown in fat pads of mice as described. BOCTAG-T2 and mock tumours were grown in the same mouse and treated systemically by intraperitoneal (i.p.) administration for 5 days with 200 mg/kg Ac$_4$GalN6yne (*n* = 3), Ac$_4$ManNAlk (*n* = 2) or the corresponding volume of vehicle (*n* = 1). Tumours were harvested, lysed, subjected to CuAAC with biotin-picolyl azide and analysed by streptavidin blot. Hex = Hexose, e.g. galactose; NeuAc = *N*-acetylneuraminic acid; HexNAc = *N*-acetylhexosamine, e.g. GlcNAc. mock: pSBbi-Hyg. Source data are provided as a Source Data file.

identification. Using a tandem MS technique consisting of Higher-energy Collision Dissociation (HCD)-triggered Electron Transfer Dissociation (ETD), we identified 37 unique glycosylation sites on 57 murine glycopeptides from 4T1 cells and 9 unique glycosylation sites on 12 human glycopeptides from MCF7 cells in secretome samples (Fig. 5c, Supplementary Data 2). Our data indicated glycosylation of homologous glycopeptides from murine and human origins in pro-X carboxypeptidase in secretome (Fig. 5d, Supplementary Fig. 16). We also performed a MS-glycoproteomics experiment in lysate from the 4T1/MCF7 co-culture expressing BOCTAG-T2 or empty plasmid. We annotated a total of 4 unique glycosylation sites on 11 murine glycopeptides from 4T1 samples and 2 unique glycosylation sites on 8 human glycopeptides from MCF7 cells (Supplementary Data 3). Particularly, we identified a homologous glycopeptide from both human and murine glucosidase 2 (Supplementary Fig. 17). The presence of the chemical tag facilitated manual annotation of mass spectra in all cases due to the specific mass shift associated with the chemical modification, in line with our previous results[37].

### Bioorthogonal cell-specific tagging of glycoproteins in vivo

We next investigated the applicability of our BOCTAG strategy in an in vivo tumour model. Tumours were grown in the fat pads of NOD-SCID IL2Rgnull (NSG) mice, consisting of 4T1 cells expressing GFP and either BOCTAG-T2 (one fat pad) or no additional transgene (empty plasmid, another fat pad). These mice were intraperitoneally injected with Ac$_4$GalN6yne, vehicle or Ac$_4$ManNAlk once daily for 5 consecutive days (Fig. 5e). At the end of the treatment, the tumours were harvested, homogenised, treated with biotin-picolyl azide under CuAAC conditions, and the labelling was analysed by streptavidin blot. A strong fluorescent signal was observed in BOCTAG-T2 tumours treated with Ac$_4$GalN6yne (Fig. 5e). In contrast, tumours transfected with empty plasmid showed minimal labelling signal with either vehicle or Ac$_4$GalN6yne treatment. All samples treated with Ac$_4$ManNAlk irrespective of the presence of NahK/mut-AGX1/BH-T2 displayed strong fluorescent signal. These data demonstrated that glycoproteins are selectively tagged when NahK/mut-AGX1/BH-T2 are expressed in the tumour. We alternatively performed intratumoral injections of either Ac$_4$GalN6yne or DMSO and observed the same BOCTAG-T2-dependent labelling (Supplementary Fig. 18a).

To evaluate the protein expression levels of NahK/mut-AGX1/BH-T2 ex vivo, a portion of the tumours was digested into single cell suspensions, plated and cell cultured. Protein expression of NahK/mut-AGX1/BH-T2 was assessed by Western blot and found to be comparable to expression levels before in vivo injection (Supplementary Fig. 18b). Cells also generally retained the ability to incorporate Ac$_4$GalN6yne-dependent chemical glycoproteome tagging (Supplementary Fig. 18b).

## Discussion

We developed BOCTAG to address two major shortcomings in the biosciences. First, there is still an unmet need for characterising proteins produced by a particular cell type. Glycans are a means to an end in this respect, and the large signal-to-noise ratio in our fluorescent labelling experiments indicates that BOCTAG allows for efficient protein tagging. The approach is complementary to other techniques, including the use of unnatural amino acids, proximity biotinylation and ligand-targeting delivery approaches[38–40]. Second, directly incorporating glycans in the analysis will give insight into cell-type-specific glycosylation sites and possibly glycan structures to add another dimension to proteome profiling. Chen and colleagues have engineered the AGX1 splice form AGX2 to accept an alkyne-tagged analogue of GlcNAc, achieving cell-specific glycoproteome tagging in vivo with focus on intracellular O-GlcNAc glycans[41]. While chemical tagging of intracellular O-GlcNAc glycans by BOCTAG is likely, we have focused on incorporation into cell surface glycans, including N-linked and, by

virtue of engineered GTs, O-GalNAc glycans. The presence of a modification that can be observed by MS as a direct corollary of chemical tools allows for further validation of enriched glycoproteins, facilitating glycoproteome analysis even in complex co-culture or in vivo settings. An artificial biosynthetic pathway was essential to ensure minimal background labelling while being able to supply the tagged sugar as an easy-to-synthesise MOE reagent. To this end, the use of the kinase NahK allows for use of a per-acetylated bioorthogonal sugar that is fundamental to in vivo use and in marked difference to highly unstable caged sugar-1-phosphates used previously[19,24]. To enable BOCTAG, cells require transfection with at least two transgenes. However, the design of a multicistronic, transposase-responsive plasmid ensures that transfection efforts are straightforward[42,43]. BOCTAG allowed us to selectively tag tumour glycoproteomes in vivo, highlighting the robustness of the approach. MOE reagents have been chemically caged to be released by enzymes overexpressed in cancer[44–46]. While independent of transfection, such targeting can be accompanied by substantial background labelling in non-cancerous tissue. BOCTAG allows for programmable glycoprotein tagging with remarkable signal-to-noise ratio, and is an enabling technology that will transform our understanding of tumour-host interactions particularly in the context of protein glycosylation.

## Methods

### Materials

The compounds Ac$_4$ManNAlk and Ac$_4$GalNAlk were purchased from Click Chemistry Tools (Scottsdale, USA). GalNAlk, GalN6yne, GalNAlk-1-phosphate, Ac$_3$GalN6yne-1-P(SATE)$_2$ and Ac$_4$GalN6yne were prepared as reported[18,23,24]. Primers were from Merck KGaA (Darmstadt, Germany). Restriction enzymes were from New England Biolabs. The plasmids pSBtet-GH and pSBbi-GH were a gift from Eric Kowarz (Addgene plasmid #60498; RRI- D:Addgene_60498, and Addgene plasmid # 60514; RRID:Addgene_60514)[43]. pTriEx6-His-GST-3C-MCS was an in-house construct modified from pTriEx-6 (Supplementary Data 4)[19]. The plasmid pCMV(CAT)T7-SB100 (Addgene plasmid #34879; Addgene: pCMV(CAT)T7-SB100; RRI- D:Addgene_34879) was a kind gift from Zsuzsanna Izsvak[42].

All sequences of oligonucleotides were reported in Supplementary Data 4.

Adobe Illustrator 2022 was used to assemble figures and to re-label plot axes.

### Cloning and expression of GALK1 and GALK2

pDONR221 plasmids containing GALK1 or GALK2 were purchased from DNASU Plasmid Repository (Arizona State University, Tempe, USA). The coding sequences in pDONR221 (HsCD00746245 for GALK1 and HsCD00039428 for GALK2) were cloned into BamHI/BglII-digested pTriEX with an N-terminal GST-3c cleavage site and a C-terminal FLAG-tag by using the In-Fusion HD Cloning Kit (Takara, Tokyo, Japan) and the following primers: inf_GALK1_F/R or inf_GALK2_F/R. Reaction was run for 30 cycles using the following protocol: 98 °C (30 s), 98 °C (10 s), 58 °C (15 s), 72 °C (3 min – return to step 2), 72 °C (5 min). Sf21 cells (ATCC CRL-1711) were transfected with pTriEx6-His-GST-3C-GALK1-FLAG and pTriEx6-His-GST-3C-GALK2-FLAG constructs using the Baculovirus system with a transfer plasmid method. Sf21 cells were infected for 3 days with recombinant baculovirus generated based on the *flash*BAC™ system (Oxford Expression Technologies, Oxford, UK). Cells were harvested (5 min, 2000 *g*, room temperature (RT)) and resuspended in 50 mL cold Lysis Buffer (50 mM HEPES pH 7.5, 150 mM NaCl, 1 mM EDTA, 1 mM DTT) with cOmplete protease inhibitors (Roche, Penzberg, Germany) and BaseMuncher mix (1:10,000, Expedeon, Cambridge, UK), and left at 4 °C for 1 h. Cells were then lysed by 6 min sonication (Duty cycle: 50%, Output control: 5–6) using a Sonifier 450 (Branson, Hampton, USA) prior to ultra-centrifugation (30000 *rpm*, 30 min). The supernatant was collected and incubated

overnight with 2 mL of GST-4B Sepharose slurry beads (Sigma Aldrich, St. Louis, USA) pre-equilibrated in cold Lysis Buffer containing 10% (v/v) glycerol. The supernatant was then collected (2000 g, 3 min, 4 °C) and washed twice with 10 column volumes (CV) of same buffer. GST-4B Sepharose beads were then resuspended in 2 CV of Lysis Buffer containing 10% (v/v) glycerol and incubated with 100 µL of HRV 3 C protease (4 mg/mL, produced in-house)[47] at 4 °C for 4 h. The supernatant was collected and the digestion was repeated three times. Supernatants were pooled and concentrated to 2 mL using a 30 kDa MWCO Amicon® Ultra-15mL Centrifugal Filter Unit(Merck). The concentrated sample was injected onto an ÄKTA™ Pure system, running a Superdex™ S200 16/600 gel filtration column (GE Life Sciences, Marlborough, USA), collecting 1 mL fractions in Lysis Buffer containing 10% (v/v) glycerol. Fractions were pooled, concentrated using a 30 kDa MWCO Amicon® Ultra-15mL Centrifugal Filter Unit, concentration measured by Nanodrop® (1.94 mg/mL), sample diluted twofold in a freezing buffer (25 mM HEPES pH 7.5, 40% (v/v) glycerol, 1 mM DTT) and stored at −80 °C until use.

## Cloning and expression of WT- and mut-AGX1

A pIRES-puro3 plasmid containing WT-AGX1 was used as template to generate mut-AGX1 by site-directed mutagenesis by using the primers AGX1_F383A_F/R and the Q5®-site direct Mutagenesis Kit[24]. WT- and mut-AGX1 fragments were cloned into BamHI/BglII-digested pTriEX with an N-terminal GST-3c cleavage site and a C-terminal FLAG-tag by using In-Fusion HD Cloning Kit (Takara, Tokyo, Japan) and primers inf_AGX1_F/R. The reaction was run for 30 cycles using the following protocol: 98 °C (30 s), 98 °C (10 s), 58 °C (15 s), 72 °C (3 min – return to step 2), 72 °C (5 min). pTriEx6-His-GST-3C-WT-AGX1-FLAG and pTriEx6-His-GST-3C-mut-AGX1-FLAG were transfected and expressed in Sf21 cells by following the same procedure described above for pTriEx6-His-GST-3C-GALK1/2-FLAG.

## In vitro phosphorylation of GalN6yne and GalNAlk

*B. Longum*, *T. bernardi*, *A. phocae*, C. sp FS41 and *E. tayi* NahKs were produced by Prozomix Ltd. (Haltwhistle, UK)[28]. Reactions with NahK enzymes or human kinases GALK1 and GALK2 were run in 50 µL volume, containing GalN6yne or GalNAlk (5 mM), adenosine triphosphate (ATP, 10 mM), MgCl$_2$ (10 mM), Tris-HCl pH 8 (100 mM) and kinase (20 µg). Reactions were run for 4 h at 37 °C. Reactions were diluted with 50 µL methanol, cooled to −20 °C for 2 h and centrifuged (18,000 g, 30 min) to remove any precipitated enzyme. Supernatants were analysed by Ultra Performance Liquid Chromatography-Mass Spectrometry (UPLC-MS) equipped with a ACQUITY UPLC BEH Glycan (Waters Corp., Milford, USA) 1.7 µm 2.1 × 50 mm column (90−65% buffer B over 17 min; buffer A: 10 mM ammonium formate pH 4.5, buffer B: 10 mM ammonium formate 90/10 (v/v) acetonitrile/water). Estimated conversion was obtained by analysing a 5 mM product standard (GalN6yne-1-phosphate or GalNAlk-1-phosphate) in the same analysis conditions as above, extracting product mass and comparing ion count with the ion count of the extracted mass of product in the samples.

## In vitro synthesis of UDP-GalN6yne or UDP-GalNAlk

In vitro enzymatic assays were run in 15 µL volume, containing GalNAlk- or GalN6yne-1-phosphate (2.5 mM, Supplementary Methods), MgCl$_2$ (5 mM), Tris-HCl pH 8 (75 mM), BSA (1 mM), uridine triphosphate (UTP, 5 mM), pyrophosphatase (PmPpA, 0.045 U, Chemily Glycoscience, Peachtree Corners, USA), recombinant WT- or mut-AGX1 (125 nM). Reactions were run for either 2 h or 16 h at 37 °C. 7 µL of each reaction were diluted with 7 µL of acetonitrile, cooled on ice for 30 min and centrifuged (18,000 g, 30 min) to remove any precipitated enzyme. Supernatants were analysed by UPLC-MS equipped with an ACQUITY UPLC BEH Glycan 1.7 µm 2.1 × 50 mm column (90−65% buffer B over 17 min; buffer A: 10 mM ammonium formate pH 4.5, buffer B: 10 mM

ammonium formate 90/10 (v/v) acetonitrile/water). The percentage conversions of sugar-1-phosphate analogues to corresponding UDP-sugars was estimated by UV peak integration of products at 260 nm. The corresponding product peak areas were converted to concentration using standard curves for each of the products. The standard curves were constructed by running commercial (UDP-GalNAc) and synthetic (UDP-GalNAlk and UDP-GalN6yne) samples on UPLC, and plotting integrated area against concentration.

## Cloning of AGX1 and NahK constructs into pSBbi plasmids

pSBbi-AGX1$^{WT}$ and pSBbi-AGX1$^{F383A}$ were cloned previously by GeneArt (Thermo Fisher, Waltham, USA)[18]; NahK from *Bifidobacterium longum* (E8MF12) was cloned into pSBbi-AGX1$^{WT}$and pSBbi-AGX1$^{F383A}$ plasmids by GeneArt, containing 2A self-cleaving peptides and a C-terminal HA tag, to give the plasmids pSBbi-AGX1$^{WT}$-NahK and pSBbi-AGX1$^{F383A}$-NahK.

## Cloning of GalNAc-T1/2 constructs into pSBbi and pSBtet plasmids

WT- or "Bump-and-Hole" (BH) engineered versions of GalNAc-T1 and T2 were cloned previously from pSBtet-plasmids[24] into pSBbi-AGX1$^{WT}$-NahK or pSBbi-AGX1$^{F383A}$-NahK using an SfiI cloning strategy with primers inf_GalNAc-T1_F/R and inf_GalNAc-T2_F/R. This cloning strategy gave the following plasmids: pSBbi-AGX1$^{WT}$-NahK-T1$^{BH}$, pSBbi-AGX1$^{F383A}$-NahK-T1$^{WT}$, pSBbi-AGX1$^{F383A}$-NahK-T1$^{BH}$, pSBbi-AGX1$^{WT}$-NahK-T2$^{BH}$, pSBbi-AGX1$^{F383A}$-NahK-T2$^{WT}$, pSBbi-AGX1$^{F383A}$-NahK-T2$^{BH}$. AGX1, NahK and GalNAc-T1/T2 constructs were tagged with C-terminal FLAG, HA and VSV-G tags, respectively. The full plasmid sequences are reported in Supplementary Data 4.

## Plasmids and cell lines

All cells were screened for contamination by mycoplasma and other cell lines by the Crick Cell Services Science Technology Platform. K-562 cells (ATCC CCL-243) were propagated in RPMI (Thermo Fisher) with 10% (v/v) FBS, penicillin (100 U/mL) and streptomycin (100 µg/mL). 4T1 (GFP-expressing, ATCC CRL-2539), murine MLg fibroblast (ATCC CCL-206) and MCF7 human breast cancer cells (ATCC HTB-22) were maintained in DMEM (Thermo Fisher) with 10% (v/v) FBS, penicillin (100 U/mL) and streptomycin (100 µg/mL).

K-562 and 4T1 (GFP) stably transfected with pSBbi-GH, pSBbi-AGX1$^{WT}$ or pSBbi-AGX1$^{F383A}$ have been prepared previously[18]. K-562 cells were stably transfected with pSBbi-AGX1-NahK, pSBbi-AGX1-NahK-T1/T2, pSBtet-AGX1$^{F383A}$-T2 constructs or empty pSBtet-GH[24] using Lipofectamine LTX (Thermo Fisher) according to the manufacturer's instructions, with a 20:1 (w/w) mixture of pSBbi and pCMV(CAT)T7-SB100 plasmid DNA. After 24 h, cells were harvested and selected in growth medium containing 150 µg/mL hygromycin B (Thermo Fisher) for 7−10 days to obtain stable cells. 4T1 (GFP), MLg and MCF7 cells were stably transfected with either pSBbi-AGX1$^{F383A}$-NahK, pSBbi-AGX1$^{F383A}$-NahK-T2$^{BH}$ or empty pSBbi-Hyg using Lipofectamine 3000 (Thermo Fisher) according to the manufacturer's specifications, with a 20:1 (w/w) mixture of pSBbi and pCMV(CAT)T7-pSB100 plasmid DNA. After 24 h, cells were harvested and selected in growth medium containing 100 µg/mL hygromycin B (Thermo Fisher) for 7−10 days to obtain stable cells.

## Analysis of nucleotide-sugar biosynthesis

K-562 cells (5,000,000) stably transfected[24] with pSBbi-GH, pSBbi-AGX1$^{WT}$, pSBbi-AGX1$^{F383A}$, pSBbi-AGX1$^{WT}$-NahK and pSBbi-AGX1$^{F383A}$-NahK were fed with 100 µM (from a 100 mM stock solution in DMSO) membrane-permeable precursor Ac$_4$GalN6yne, Ac$_4$GalNAlk, Ac$_3$GalN6yne-1-P(SATE)$_2$ or DMSO. After 16 h, cells were harvested, centrifuged (500 g, 5 min, 4 °C) and resuspended in PBS (1 mL). 0.9 mL cell suspension were transferred to O-ring tubes (1.5 mL, Thermo Fisher) and harvested. Zirconia/silica beads (0.1 mm, BioSpec, Bertlesville,

USA) were added at a volume comparable to the cell pellet volume, followed by 1:1 acetonitrile/water (1 mL). Cells were lysed using a bead beater (FastPrep-24, MP Biomedicals, Santa Ana, USA) at 6 m/s for 30 s, and the cell lysate was cooled at 4 °C for 10 min. Samples were centrifuged (14,000 $g$, 10 min, 4 °C), and the supernatant was transferred to a fresh tube. The solvent was evaporated by SpeedVac, and the residue was dissolved in Milli-Q water (MQ, 0.3 mL) containing 15 μM adenosyl diphosphate (ADP)-α-D-glucose (Sigma Aldrich, St. Louis, USA). The solution was passed through a centrifuge filter (30 min, 14,000 $g$) using a 3 kDa MWCO Amicon® Ultra-0.5 mL Centrifugal Filter Unit. The flow-through was evaporated by SpeedVac and the residue resuspended in 60 μL of MQ water. High Performance Anion Exchange Chromatography (HPAEC) was used to analyse lysates.

HPAEC was carried out using an ICS-6000 equipped with a quaternary pump and a conductivity detector (data collection rate 5.0 Hz, cell temperature 35 °C) on an AS11 2 × 250 mm column and a 2 × 50 mm guard column (Thermo Fisher). Solvents were: A = water; B = 1 M NaOH. The gradient profile (0.25 mL/min flow rate) was as follows: 0 min 99.9% A, 0.1% B; 3 min 99.9% A, 0.1% B; 8 min 96.9% A, 3.1% B; 13 min 96.4% A, 3.6% B; 38 min 93% A, 7% B; 39 min 90% A, 10% B; 43 min 90% A, 10% B; 48 min 90% A, 10% B. Commercial or synthetic UDP-sugar standards (200–500 μM) were used as controls.

### Quantification of nucleotide-sugar levels

K-562 cells (6,000,000) stably transfected with pSBbi-GH, pSBbi-AGX1[WT], pSBbi-AGX1[F383A], and pSBbi-AGX1[F383A]-NahK were fed with 10 μM (from a 50 mM stock solution in DMSO) membrane-permeable precursor Ac$_4$GalN6yne or DMSO. After 16 h, cells were harvested, centrifuged (500 $g$, 5 min, 4 °C) and resuspended in PBS (1 mL). 0.9 mL cell suspension were transferred to O-ring tubes (1.5 mL, Thermo Fisher) and harvested. Zirconia/silica beads (0.1 mm, BioSpec, Bartlesville, USA) were added at a volume comparable to the cell pellet volume, followed by 1:1 acetonitrile/water (1 mL). Cells were lysed using a bead beater (FastPrep-24, MP Biomedicals, Santa Ana, USA) at 6 m/s for 30 s, and the cell lysate was cooled at 4 °C for 10 min. Samples were centrifuged (14,000 $g$, 10 min, 4 °C), and the supernatant was transferred to a fresh tube. The solvent was evaporated by SpeedVac, and the residue was dissolved in Milli-Q water (MQ, 0.3 mL) containing 15 μM ADP-α-D-glucose. The solution was passed through a centrifuge filter (30 min, 14,000 $g$) using a 3 kDa MWCO Amicon® Ultra-0.5mL Centrifugal Filter Unit. The flow-through was evaporated by SpeedVac and the residue resuspended in 60 μL of MQ water. High Performance Ion-pair Reversed-phase Chromatography (IP RP HPLC) was used to analyse lysates.

IP RP HPLC was carried out using an Agilent 1260 Infinity HPLC equipped with a Diode Array Detector (G4212B, data collection at 260 nm) on a Kinetex 5 μm EVO C18 100 Å 250 × 4.6 mm column (Phenomenex, Macclesfield, UK). Solvents were: A = 38.1 mM K$_2$HPO$_4$, 61.8 mM KH$_2$PO$_4$, 8mM (CH$_3$)$_4$N(HSO$_4$); B = 80% buffer A, 20% Acetonitrile. The gradient profile (0.3 mL/min flow rate) was as follows: 0 min 100% A; 56 min 40% A; 64 min 100% A; 78 min 0% A; 103 min 0% A. Commercial or synthetic UDP-sugar standards (1.25–50 μM) were used as controls.

### Metabolic labelling, click reaction of glycoproteins on the cell surface, and in-gel fluorescence

K-562 cells stably transfected with pSBbi-based plasmids were seeded at a density of 250,000 cells/mL into 6-well plates in growth medium without hygromycin. Cells were treated with Ac$_4$GalN6yne or Ac$_4$ManNAlk at the indicated concentrations or a corresponding volume of DMSO. Cells were grown for 20 h. Cells were harvested (500 $g$, 5 min, 4 °C) in a V-shaped 96-well plate and washed twice with 2% (v/v) FBS in PBS (Labelling Buffer, 200 μL). Cells were resuspended in Labelling Buffer (35 μL), treated with a solution of 200 μM CuSO$_4$, 1.2 mM BTTAA (Click Chemistry Tools, Scottsdale, USA), 5 mM sodium

ascorbate, 5 mM aminoguanidinium chloride and 200 μM CF680-picolyl azide (Biotium, Fremont, USA) in Labelling Buffer (35 μL), and incubated for 7 min at RT on an orbital shaker. The click reaction was quenched with 3 mM bathocuproinedisulfonic acid (BCS) in PBS (35 μL). Cells were centrifuged (500 $g$, 3 min, 4 °C), washed twice with Labelling Buffer and then with PBS, and treated with 100 μL of ice-cold Lysis Buffer (50 mM Tris-HCl pH 8, 150 mM NaCl, 1% (v/v) Triton X-100, 0.5% (v/v) sodium deoxycholate, 0.1% (w/v) Sodium Dodecyl Sulfate (SDS), 1 mM MgCl$_2$, and 100 mU/μL benzonase (Merck)) containing halt protease inhibitors (ThermoFisher). Cells were lysed for 20 min at 4 °C on an orbital shaker and centrifuged (1500 $g$, 20 min, 4 °C). Supernatant was transferred to a new plate and Pierce™ BCA Protein Assay kit (Thermo Fisher) was used to measure protein concentration. Loading buffer (a 1:1:1:0.5 (v/v/v/v) mixture of 1 M Tris-HCl pH 6.5, 80% (v/v) glycerol, 10% (w/v) SDS and 1 M dithiothreitol (DTT)) was added; samples were run on a 10% or 4–20% Criterion™ TGX™ Precast gel (Bio-Rad, Hercules, USA) for SDS-PAGE, and imaged on an Odyssey CLx imager (LI-COR Biosciences, Lincoln, USA). For protein quantification by densitometry, the signal in each lane was integrated in the 700 nm channel using Image Studio Pro software (LI-COR Biosciences, Lincoln, USA). Total protein was stained with Coomassie using Acquastain (Bulldog Bio, Portsmouth, USA). Protein expression was assessed by Western blot, on a nitrocellulose membrane, with a different aliquot of the same set of samples, using antibodies against FLAG tag (rabbit anti-FLAG antibody, 1:1000, PA1-984B, Invitrogen, Carlsbad, USA), HA tag (rabbit anti-HA antibody, 1:1000, ab9110, Abcam), VSV-G tag (goat anti-VSV-G, 1:2000, ab3861, Abcam) and GADPH (rabbit anti-GAPDH, 1:5000, ab181602, Abcam).

For enzymatic treatment, lysates from cell surface-labelled cells (5 μg protein) were diluted to 20 μL with 50 mM Tris-HCl pH 7.5 and 150 mM NaCl. Samples were either left untreated, or treated with 2 μL of a 1:10 dilution in PBS of commercial PNGase F (Promega, Madison, USA), 2 μL StcE (96.5 μg/mL solution in PBS)[35] or 2 μL of a mixture of SialEXO and OpeRATOR (4 U/μL in PBS, Genovis, Lund, Sweden). Samples were incubated for 2 h at 37 °C, briefly heated to 95 °C and cooled on ice. SDS-PAGE and in-gel fluorescence were performed as described above.

In-gel fluorescence and Western blot images were visualised and processed on Image Studio Lite software (LI-COR Biosciences, Lincoln, USA), cropped by Photoshop 2020 and assembled by Illustrator 2022 (Adobe, San Jose, USA).

### Metabolic labelling, click reaction of glycoproteins both on the cell surface and in cell lysate, and streptavidin blot

K-562 cells stably transfected with pSBbi-based plasmids were seeded at a density of 250,000 cells/mL into 6-well plates in growth medium without hygromycin. Cells were treated with Ac$_4$GalN6yne at the indicated concentrations or a corresponding volume of DMSO. Cells were grown for 20 h. Cells were harvested (500 $g$, 5 min, 4 °C) in a V-shaped 96-well plate and washed twice with 2% (v/v) FBS in PBS (Labelling Buffer, 200 μL). Cells were resuspended in Labelling Buffer (35 μL), treated with a solution of 200 μM CuSO$_4$, 1.2 mM BTTAA, 5 mM sodium ascorbate, 5 mM aminoguanidinium chloride and 200 μM biotin-picolyl azide (Sigma Aldrich) in Labelling Buffer (35 μL), and incubated for 7 min at RT on an orbital shaker. The click reaction was quenched with 3 mM BCS in PBS (35 μL). Cells were centrifuged, washed three times with PBS, and treated with 100 μL of ice-cold Lysis Buffer (50 mM Tris-HCl pH 8, 150 mM NaCl, 1% (v/v) Triton X-100, 0.5% (v/v) sodium deoxycholate, 0.1% (w/v) SDS, 1 mM MgCl$_2$, and 100 mU/μL benzonase) containing halt protease inhibitors and 50 μM of PUGNAc (Sigma Aldrich). Cells were lysed for 20 min at 4 °C on an orbital shaker and centrifuged (1500 $g$, 20 min, 4 °C). Supernatant was transferred to a new plate and Pierce™ BCA Protein Assay kit was used to measure protein concentration. 40 μg of cell surface labelled proteins were divided in two aliquots of 20 μg each. 20 μg were directly loaded

on the gel, while 20 μg were diluted to 17 μL with PBS and 3 μL of 10X CuAAC solution (3 mM CuSO₄, 6 mM BTTAA, 1 mM biotin-picolyl azide, 50 mM sodium ascorbate and 50 mM aminoguanidinium chloride) were added. The click reaction was left 6 h at RT under shaking.

Loading buffer (a 1:1:1:0.5 (v/v/v/v) mixture of 1 M Tris-HCl pH 6.5, 80% (v/v) glycerol, 10% (w/v) SDS and 1 M DTT) was added; samples were incubated at 95 °C for 5 min and run on a 10% or 4-20% Criterion™ TGX™ Precast gel. After transferring proteins on a nitrocellulose membrane, the total protein amount was compared using the REVERT protein staining kit (LI-COR Biosciences) and biotinylation detected using IRDye 800CW Streptavidin (LI-COR) according to the manufacturer's instructions.

Western blot images were visualised and processed on Image Studio Lite software (LI-COR), cropped by Photoshop 2020 and assembled by Illustrator 2022.

### Metabolic labelling, click reaction in cell lysate, and streptavidin blot

K-562 cells stably transfected with pSBbi-based plasmids were seeded at a density of 250,000 cells/mL into 6-well plates in growth medium without hygromycin. Cells were treated with Ac₄GalN6yne at the indicated concentrations or a corresponding volume of DMSO. Cells were grown for 20 h. Cells were harvested (500 $g$, 5 min, 4 °C) in a V-shaped 96-well plate and washed three times with PBS. Cells were treated with 100 μL of ice-cold Lysis Buffer (50 mM Tris-HCl pH 8, 150 mM NaCl, 1% (v/v) Triton X-100, 0.5% (v/v) sodium deoxycholate, 0.1% (w/v) SDS, 1 mM MgCl₂, and 100 mU/μL benzonase) containing halt protease inhibitors and 50 μM of PUGNAc. Cells were lysed for 20 min at 4 °C on an orbital shaker and centrifuged (1500 $g$, 20 min, 4 °C). Supernatant was transferred to a new plate and Pierce™ BCA Protein Assay kit was used to measure protein concentration.

20 μg of proteins were diluted to 17 μL with PBS and 3 μL of 10X CuAAC solution (3 mM CuSO₄, 6 mM BTTAA, 1 mM biotin-picolyl azide, 50 mM sodium ascorbate and 50 mM aminoguanidinium chloride) were added. The click reaction was left 6 h at RT under shaking.

Loading buffer (a 1:1:1:0.5 (v/v/v/v) mixture of 1 M Tris-HCl pH 6.5, 80% (v/v) glycerol, 10% (w/v) SDS and 1 M DTT) was added; samples were incubated at 95 °C for 5 min and run on a 10% or 4−20% Criterion™ TGX™ Precast gel. After transferring proteins on nitrocellulose membrane, the total protein amount was assessed using the REVERT protein staining kit and biotinylation detected using IRDye 800CW Streptavidin (LI-COR) according to the manufacturer's instructions.

Western blot images were visualised and processed on Image Studio Lite software (LI-COR) cropped by Photoshop 2020 and assembled by Illustrator 2022.

### Glycome analysis by lectin blot and immunoblot analysis

K-562 cells stably transfected with pSBbi-based plasmids were seeded at a density of 250,000 cells/mL into 6-well plates in growth medium without hygromycin. Cells were treated with Ac₄GalN6yne at 10 μM or a corresponding volume of DMSO. Cells were grown for 20 h. Cells were harvested (500 $g$, 5 min, 4 °C) in a V-shaped 96-well plate and washed three times with PBS. Cells were treated with 100 μL of ice-cold Lysis Buffer (50 mM Tris-HCl pH 8, 150 mM NaCl, 1% (v/v) Triton X-100, 0.5% (v/v) sodium deoxycholate, 0.1% (w/v) SDS, 1 mM MgCl₂, and 100 mU/μL of PUGNAc) containing halt protease inhibitors and 50 μM of PUGNAc. Cells were lysed for 20 min at 4 °C on an orbital shaker and centrifuged (1500 $g$, 20 min, 4 °C). Supernatant was transferred to a new plate and Pierce™ BCA Protein Assay kit was used to measure protein concentration.

20 μg of proteins were diluted to 17 μL with PBS and loading buffer (a 1:1:1:0.5 (v/v/v/v) mixture of 1 M Tris-HCl pH 6.5, 80% (v/v) glycerol, 10% (w/v) SDS and 1 M DTT) was added; samples were incubated at

95 °C for 5 min and run on a 10% or 4-20% Criterion™ TGX™ Precast gel. After transferring proteins on a nitrocellulose membrane, the total protein amount was assessed using the REVERT protein staining kit. The membrane was blocked 30 min at RT with Intercept (TBS) Blocking buffer (LI-COR).

O-GlcNAc-containing glycoproteins were detected by overnight staining at 4 °C with mouse anti-RL2 (1:500, ab2739, Abcam) in Intercept (TBS) Blocking buffer (LI-COR) followed by anti-mouse secondary antibody according to the manufacturer's instructions.

Prior to blot staining with biotinylated lectin, endogenous biotinylated proteins were blocked by using the Endogenous Avidin/Biotin blocking kit (ab64212, Abcam) according to the manufacturer's instructions. To assess the glycan distribution, 1 h incubation in 0.05% (v/v) Tween-20 (PBS) at RT with 20 μg/mL of the following biotinylated lectins was performed: Concanavalin A (ConA, B-1105, Vector Labs, Newark, USA), Aleuria Aurantia Mushrooms (AAL, B-1395-1, Vector Labs), Maackis Amurensis II (MAL II, B-1265-1, Vector Labs), Sambucus Nigra (SNA, B-1305-2, Vector Labs). ConA staining required 0.1 mM of CaCl₂ in 0.05% (v/v) Tween-20 (PBS) incubation buffer.

The binding of biotinylated lectins was detected by using IRDye 800CW Streptavidin (LI-COR) according to the manufacturer's instructions.

Blot images were visualised and processed on Image Studio Lite software (LI-COR) cropped by Photoshop 2020 and assembled by Illustrator 2022.

### Induction of GalNAc-T2 expression and lectin blot

K-562 cells stably transfected with pSBtet-GH and pSBtet-AGX1^F383A-T2 constructs were seeded at a density of 250,000 cells/mL into 6-well plates in growth medium without hygromycin. Cells were treated with different concentration of sterile-filtered doxycycline or a corresponding volume of PBS. After 24 h, media was replaced with fresh doxycycline-containing media and incubated for further 24 h. Cells were harvested (500 $g$, 5 min, 4 °C) in a 96-well plate and washed twice with PBS. Cells were treated with 100 μL of ice-cold Lysis Buffer (50 mM Tris-HCl pH 8, 150 mM NaCl, 1% (v/v) Triton X-100, 0.5% (v/v) sodium deoxycholate, 0.1% (w/v) SDS, 1 mM MgCl₂, and 100 mU/μL benzonase) containing halt protease inhibitors and 50 μM of PUGNAc. Cells were lysed for 20 min at 4 °C on an orbital shaker and centrifuged (1500 $g$, 20 min, 4 °C). Supernatant was transferred to a new plate and Pierce™ BCA Protein Assay kit was used to measure protein concentration.

20 μg of proteins were diluted to 17 μL with PBS and loading buffer (a 1:1:1:0.5 (v/v/v/v) mixture of 1 M Tris-HCl pH 6.5, 80% (v/v) glycerol, 10% (w/v) SDS and 1 M DTT was added; samples were incubated at 95 °C for 5 min and run on a 4−20% Criterion™ TGX™ Precast gel. After transferring proteins on nitrocellulose membrane, the total protein amount was assessed using the REVERT protein staining kit and GalNAc-T2 expression levels were determined by 3 h incubation at RT with rabbit anti-VSV-G (1:500, PA129903, Invitrogen) followed by anti-rabbit secondary antibody according to the manufacturer's instructions. MAL II binding was determined as described in the section above. Blot images were visualised and processed on Image Studio Lite software (LI-COR) cropped by Photoshop 2020 and assembled by Illustrator 2022.

### Transcriptome analysis

K-562 cells (1,000,000), transfected with pSBbi-GH or pSBbi-AGX1^F383A-NahK, were either collected for RNA extraction (unfed) or fed with DMSO, 10 μM Ac₄GalN6yne or 10 μM Ac₄GalNAc. After overnight feeding (16 h), cells were collected for RNA extraction. Unfed and fed pellets were collected on different days. Biological replicates of each sample were collected on the same day.

RNA was extracted from fed and unfed cells by using RNeasy Mini kit (Qiagen, Crawley, UK), DNase I (Thermo Fisher) and QIAshredder (Qiagen) according to the manufacturer's instructions. mRNA capture and library preparation were performed by the Advanced Sequencing Facility at the Francis Crick Institute using the KAPA mRNA HyperPrep Kit (Roche, Basel, Switzerland). Technical triplicate libraries were sequenced on an Illumina HiSeq 4000 platform at the facility to generating on average 12 million 101 bp single-end reads per sample.

Raw reads were quality and adapter trimmed using cutadapt (version 1.5)[48] before alignment. Reads were mapped and subsequent gene-level counted using RSEM 1.3.0[49] and STAR 2.5.2[50] against the human genome GRCh38 using annotation release 86, both from Ensembl. Normalisation of raw count data and differential expression analysis was performed with the DESeq2 package (version 1.24.0)[51] within the R programming environment (version 3.6.1)[52]. The following pairwise comparison were performed: pSBbi-AGX1$^{F383A}$-NahK samples vs pSBbi-GH samples; pSBbi-AGX1$^{F383A}$-NahK-DMSO-fed samples vs GH-pSBbi-DMSO-fed samples; pSBbi-AGX1$^{F383A}$-NahK-GalNAc-fed samples vs pSBbi-GH- GalNAc-fed samples; pSBbi-AGX1$^{F383A}$-NahK-GalN6yne-fed samples vs pSBbi-GH-GalN6yne-fed samples; and a likelihood ratio test analysis for each cell type across all feeding regimes with the contrast function, from which genes differentially expressed (adjusted $p$ value being <0.05) between different conditions were determined. Gene lists were used to look for pathways, biological processes, cellular components and molecular functions enrichment using the Broad's GSEA software (version 3.0) with genesets from MSigDB (version 7.1)[53].

## SILAC-based quantitative proteome analysis

K-562 cells stably transfected with pSBbi-GH and pSBbi-AGX1$^{F383A}$-NahK (or pSBbi-AGX1$^{F383A}$-NahK-BH-GalNAc-T2) were individually grown in *heavy* and *light* media for 6 doublings, sufficient to achieve a labelling efficiency of >95%, before being fed with either DMSO or 10 μM Ac$_4$GalN6yne. For each feeding sample, 5,000,000 cells in 30 mL of feeding media were used.

Light media contained RPMI (Thermo Fisher) with 10% (v/v) dialysed FBS, proline (0.1 mg/mL. Thermo Fisher) and $^{12}$C/$^{14}$N light lysine/arginine K0/R0 (0.1 mg/mL, Thermo Fisher). Heavy media contained RPMI (Thermo Fisher) with 10% (v/v) dialysed FBS, proline (0.1 mg/mL) and $^{13}$C$_6$, $^{15}$N$_2$ heavy lysine/arginine K8/R10 (0.1 mg/mL, Thermo Fisher) that replaced natural lysine and arginine.

After 20 h, cells were centrifuged (500 $g$, 5 min) and washed twice with PBS (200 μL). After being transferred into a V-shaped 96-well plate, cells were lysed with 200 μL of ice-cold Lysis Buffer (50 mM Tris-HCl pH 8, 150 mM NaCl, 1% (v/v) Triton X-100, 0.5% (v/v) sodium deoxycholate, 0.1% (w/v) SDS, 1 mM MgCl$_2$, and 100 mU/μL benzonase) containing 50 μM PUGNAc. Cells were lysed for 20 min at 4 °C on an orbital shaker and centrifuged (1500 $g$, 20 min, 4 °C). Supernatant was transferred to a new plate and Pierce™ BCA Protein Assay kit was used to measure protein concentration.

Heavy and light lysates were mixed 1:1 (0.25 mg each), normalised up to 250 μL with PBS and incubated for 2 h at RT with 300 μL of Neutravidin beads slurry (Sera-Mag SpeedBeads Neutravidin-Coated Magnetic Beads, cytiva, Marlborough, USA), previously washed with PBS (2 × 200 μL), to remove endogenous biotinylated proteins. The supernatant was collected, diluted to 270 μL with PBS and 30 μL of 10X CuAAC solution (3 mM CuSO$_4$, 6 mM BTTAA, 1 mM biotin-picolyl azide, 50 mM sodium ascorbate and 50 mM aminoguanidinium chloride) were added. The click reaction was left 6 h at RT under shaking. Samples were treated with 3 mL cold methanol (−20 °C, tenfold excess) and left 24 h at −80 °C for protein precipitation.

Samples were then centrifuged (3700 $g$, 4 °C, 20 min) and supernatant discarded; pellets were washed twice with cold methanol (3 mL) and centrifuged between washes (3700 $g$, 4 °C, 20 min). Supernatant was completely removed (tubes upside-down on tissue paper, then let air dry) and samples were resuspended in 250 μL 0.1% (w/v) Rapigest (Waters) in PBS and sonicated in a water bath for 25 min. Samples were centrifuged (3700 $g$, 5 min), supernatants were transferred to new tubes and pellets were treated with 250 μL of 6 M urea in PBS. Samples were sonicated for 25 min and centrifuged again (3700 $g$, 5 min). The pellets were resuspended with 250 μL of PBS, sonicated for 25 min and centrifuged again. Rapigest, urea and PBS supernatants were then combined and incubated with 350 μL of dimethylated Neutravidin Beads slurry[54] (previously washed twice with 200 μL of PBS) for 2 h at RT. Supernatant was discarded and beads were washed with 1% (w/v) Rapigest (3 × 350 μL), 6 M urea in PBS (6 × 350 μL), AmBic (50 mM ammonium bicarbonate, 6 × 350 μL) and 40% (v/v) LCMS-grade acetonitrile (4 × 100 μL). Beads were resuspended in 100 μL of AmBic containing 10 mM DTT and then incubated at 50 °C for 15 min. Beads were washed with AmBic (2 × 350 μL) and 100 μL of 20 mM iodoacetamide in AmBic was then added. Samples were kept for 30 min in the dark. Iodoacetamide was then quenched by adding DTT 10 mM (final concentration). The beads were washed with AmBic (3 × 350 μL), then resuspended in 100 μL of AmBic and 300 ng of Lys-C Mass Spec Grade (Promega) were added to beads followed by overnight incubation at 37 °C. The supernatant was transferred to a new tube and 200 ng of Trypsin gold Mass Spec Grade (Promega) were added. The digestion was left for 8 h at 37 °C. Peptides were desalted by UltraMicroSpin™ (The Nest group Inc., Ipswich, USA) according to the manufacturer's protocol and vacuum-dried by SpeedVac to remove any traces of organic solvents.

Dried peptides were resuspended in 16 μL of 0.1% (v/v) formic acid in LCMS-grade water, sonicated for 15 min in a water bath, vortexed briefly and harvested 5 min at 18,000 $g$. Peptide mixtures were analysed by nanoflow LC-MS/MS using an Orbitrap Fusion Lumos with ETD (Electron Transfer Dissociation, Thermo Fisher) coupled to an Ulti-Mate 3000 RSLCnano (Thermo Fisher). The samples (15 μL) were loaded via autosampler isocratically onto a 50 cm, 75 μm PepMap RSLC C18 column after pre-concentration onto a 2 cm, 75 μm Acclaim Pep-Map100 m nanoViper.

The column was held at 40 °C using a column heater in the EASY-Spray ionization source (Thermo Fisher). The samples were eluted at a constant flow rate of 0.275 μL/min using a 240 min gradient. Solvents were: A = 5% (v/v) DMSO, 95% (v/v) 0.1% formic acid in water; B = 5% (v/v) DMSO, 20% (v/v) 0.1% formic acid in water, 75% (v/v) 0.1% formic acid in acetonitrile.

The gradient profile was as follow: 0 min 98% A, 2% B; 5 min 98% A, 2% B; 55 min 98% A, 2% B; 190 min 70% A, 30% B; 213 min 60% A, 40% B; 213 min 5% A, 85% B; 213 min 5% A, 85% B; 225 min 98% A, 2% B: 240 min 98% A, 2% B.

MS1 scans were collected with a mass range from 350 to 1500 $m/z$, 120 K resolution, $4 × 10^5$ ion inject target, and 50 ms maximum inject time. Dynamic exclusion was set to exclude for 45 s with a repeat count of 1. Charge states 2–6 with an intensity >$1 × 10^4$ were selected for fragmentation at top speed for 3 s. Selected precursors were fragmented using HCD at 30% nCE with 1.2 Da isolation window, $1 × 10^4$ inject target, and 100 ms maximum inject time. MS2 scans were taken in the ion trap at a rapid scan rate.

Raw MS files were loaded into MaxQuant software (version 1.6.5.0)[55] for quantification and identification by using Homo sapiens FASTA protein sequences database from UniProt (downloaded 18$^{th}$ June, 2020) for the database search[56]. The protein groups table were uploaded into Perseus (version 1.6.14.0)[57] to allow for data transformation and visualization, and into RStudio for statistical analysis.

Search parameters included standard group-specific parameter type with multiplicity of 2 and maximum labelled of 3. Specific cleavage specificity of R and K, with two missed cleavages were allowed. Methionine oxidation and N-terminal acetylation were set as variable modifications with a total common max of 5. Cysteine

carbamidomethylation was set as a fixed modification. Minimum peptide length was set at 7 amino acid units. Mass tolerance was set at 20 ppm for MS1s, 10 ppm for HCD MS2s. Peptide- and protein-level FDR were set at 0.01. R programming environment (version 4.1.3)[52] was used for statistical analysis. Differentially enriched proteins were determined using empirical Bayes moderated $t$-test (two-sided) by limma package. Multiple-testing corrections were performed using the Benjamini-Hochberg procedure to calculate adjusted $p$ values (False Detection Rate). Protein hits were filtered using a $p$ value of 0.1.

### Fluorescence microscopy

For monoculture samples, non-transfected and pSBbi-AGX1$^{F383A}$-NahK stably transfected 4T1 (GFP-expressing) and fibroblast MLg cells were seeded into a μ-Plate 24 Well Black (Thistle Scientific Ltd, Glasgow, UK) at a density of 30,000 cells in 350 μL growth medium (DMEM with 10% (v/v) FBS, penicillin (100 U/mL), streptomycin (100 μg/mL)) without hygromycin. Cells were treated with either DMSO, 50 μM Ac$_4$GalN6yne or 50 μM Ac$_4$ManNAlk.

For co-culture samples, non-transfected and pSBbi-AGX1$^{F383A}$-NahK stably transfected MLg cells were seeded into a μ-Plate 24 Well Black (Thistle Scientific Ltd) at a density of 30,000 cells in 350 μL growth medium without hygromycin. After 4 h, 3000 4T1 (GFP) cells were plated on the top of 30,000 MLg cells (1:10 ratio). The co-culture samples were grown 72 h before feeding with either DMSO, 50 μM Ac$_4$GalN6yne or 50 μM Ac$_4$ManNAlk.

Cells were incubated for 16 h. Medium was aspirated, and cells were washed with ice-cold 2% (v/v) FBS in PBS (2 × 200 μL). Cells were then treated with 200 μL of a freshly prepared CuAAC solution containing 300 μM BTTAA, 50 μM CuSO$_4$, 5 mM sodium ascorbate, 5 mM aminoguanidinium chloride and 200 μM biotin-picolyl azide. The reaction was carried out for 3 min at RT, the supernatant was aspirated and cells were washed with ice-cold PBS (4 × 200 μL). Cells were incubated for 20 min at RT with 20 μg/mL Streptavidin-AlexaFluor647 (BioLegend UK Ltd, Kentish Town, UK) in 1% (w/v) BSA in PBS solution in the dark. After washing with ice-cold PBS (4 × 200 μL), cells were fixed 20 min with cold 4% (v/v) formaldehyde (Thermo Fisher) in 100 mM sodium phosphate buffer pH 7.4, at RT in the dark. The reaction was quenched by 5 min incubation with 50 mM ammonium chloride and the cells were washed with PBS (3 × 200 μL). Cells were permeabilised with 0.1% (v/v) Triton X-100 in PBS for 10 min at 4 °C and washed with PBS (3 × 200 μL). Cells were blocked in a solution of 10% (v/v) normal donkey serum (ab7475, Abcam), 1% (w/v) BSA and 0.1% (v/v) Tween-20 in PBS for 1 h at RT. GFP expression was detected by incubation with goat anti-GFP (1:300, ab5450, Abcam) in a solution containing 5% (v/v) donkey serum in 1% (w/v) BSA in PBS. Cells were washed with PBS (3 × 200 μL) and incubated for 30 min at RT with AlexaFluor488 anti-goat secondary antibody (1:500, ab150129, Abcam) in a solution containing 1% (v/v) normal donkey serum in 1% (w/v) BSA in PBS. Cells were then incubated with AlexaFluor568 Phalloidin (Invitrogen A12380, 5 μL of 40X methanol stock solution in each 200 μL of PBS) in 1% (w/v) BSA and 0.1% (v/v) Tween-20 followed by PBS washing (3 × 200 μL) and DAPI incubation (1:1000, Vector Laboratories Ltd, Peterborough, UK) in 1% (w/v) BSA and 0.1% (v/v) Tween-20 in PBS for 30 min at RT. After washing with PBS (3 × 200 μL), circle CoverSlips with 15 mm diameter (Thermo Fisher) were mounted onto each well by using 20 μL of ProLong Gold Antifade Mountant (Invitrogen).

The confocal acquisition was made on a Zeiss LSM710 Invert microscope. The images were acquired using a Plan Apochromat 40×/1.3 Oil objective. Monoculture samples were imaged with an acquisition zoom of 1.3, so the corresponding resulting pixel size was 0.159 μm. For the co-culture samples, three-dimensional images were acquired with an acquisition zoom of 0.6, so the corresponding resulting pixel size was 0.219 μm (x, y) and 0.7 μm (z).

A sequential scan to spectrally separate the fluorescence of DAPI, AlexaFluor647, AlexaFluor488 and AlexaFluor568 was used. In addition, the transmitted light channel was activated to visualise the cell morphology. Images were visualised and processed with Fiji and Zen (Zeiss, Oberkochen, Germany) software[58].

### Proteome and glycoproteome analyses in co-culture samples

Murine 4T1 (GFP-expressing) and human MCF7 cells stably transfected with pSBbi-AGX1$^{F383A}$-NahK-T2$^{BH}$ and pSBbi-Hyg plasmids were individually plated or co-cultured (1:1 ratio) overnight in DMEM with 10% (v/v) FBS, penicillin (100 U/mL), streptomycin (100 μg/mL) and hygromycin B (100 μg/mL). For each mono- and co-culture sample, two 15 cm petri dishes containing a total of 6,000,000 cells were prepared.

After overnight growing, the media was discarded and replaced with fresh media containing either DMSO or 10 μM Ac$_4$GalN6yne. FBS-free media was used for feeding samples addressed to secretome glycoproteins labelling.

**Secretome glycoprotein enrichment.** FBS-free media was collected and centrifuged (500 $g$, 5 min) to remove any cell debris. The secretome was then concentrated up to 300 μL and media replaced with PBS by using a 10 kDa MWCO Amicon® Ultra-15 Centrifugal Filter Unit. Pierce™ BCA Protein Assay kit was used to measure protein concentration. 300 μg of each sample was used for the next step.

**Cell lysate glycoprotein enrichment.** Media was discarded and cells were washed with PBS (5 mL). Cells were detached from the petri dish by 10 min incubation at RT with 8 mM EDTA in PBS (10 mL) and then washed twice with PBS. Cells were lysed with 200 μL of ice-cold Lysis Buffer (50 mM Tris-HCl pH 8, 150 mM NaCl, 1% (v/v) Triton X-100, 0.5% (v/v) sodium deoxycholate, 0.1% (w/v) SDS, 1 mM MgCl$_2$, and 100 mU/μL benzonase) containing 50 μM PUGNAc. Cells were lysed for 20 min at 4 °C on an orbital shaker and centrifuged (1500 $g$, 20 min, 4 °C). Supernatant was transferred to a new plate and Pierce™ BCA Protein Assay kit was used to measure protein concentration.

For each lysate, 300 μg of protein was brought to 250 μL with PBS and incubated for 2 h at RT with 300 μL of Neutravidin beads slurry, previously washed with PBS (2 × 200 μL), to remove endogenous biotinylated proteins. Both secretome and lysate samples were diluted to 270 μL with PBS and 30 μL of 10X CuAAC solution (3 mM CuSO$_4$, 6 mM BTTAA, 1 mM DADPS-biotin-picolyl azide (Sussex Research Laboratories Inc., Ontario, CA), 50 mM sodium ascorbate and 50 mM aminoguanidinium chloride) were added. The click reaction was left 6 h at RT under shaking. Samples were treated with 3 mL cold methanol (−20 °C, tenfold excess) and left 24 h at −80 °C for protein precipitation.

Pellets were washed, resuspended and enriched on dimethylated Neutravidin Beads[54] by following the same procedure described in the *SILAC-based quantitative proteomics analysis* section. Peptides were eluted as described above.

Beads with bound glycopeptides following Lys-C on-bead digest were incubated with 150 μL of 1% (v/v) formic acid in LCMS-grade water 30 min at RT on a rotator. The supernatant was collected and the acid-cleavage treatment was repeated a second time. Beads were washed with LCMS-grade acetonitrile. The wash and the acidic supernatants were combined together and 200 ng of Trypsin gold Mass Spec Grade were added. The digestion was left 8 h at 37 °C. Glycopeptides were dried by SpeedVac and then resuspended in AmBic containing 2% (v/v) LCMS-grade acetonitrile and 0.1% (v/v) formic acid.

Peptides and glycopeptides were desalted by UltraMicroSpin™ according to the manufacturer's protocol and vacuum-dried by SpeedVac to remove any traces of organic solvents.

Dried peptides and glycopeptides were resuspended in 16 μL of 0.1% (v/v) formic acid in LCMS-grade water, sonicated for 15 min, vortexed briefly and centrifuged for 5 min at 18,000 $g$.

Sample mixtures were analysed by nanoflow LC-MS/MS using an Orbitrap Eclipse with ETD (Thermo Fisher) coupled to an UltiMate 3000 RSLCnano (Thermo Fisher).

The sample (15 µL for glycopeptide fractions and 3 injections of 5 µL each for peptide fractions) was loaded via autosampler isocratically onto a 50 cm, 75 µm PepMap RSLC C18 column (ES903) after pre-concentration onto a 2 cm, 75 µm Acclaim PepMap100 m nanoViper.

The column was held at 40 °C using a column heater in the EASY-Spray ionization source (Thermo Fisher). The samples were eluted at a constant flow rate of 0.275 µL/min using a 120 and 140 min gradient for peptides and glycopeptides, respectively. Solvents were: A = 5% (v/v) DMSO, 95% (v/v) 0.1% formic acid in water; B = 5% (v/v) DMSO, 20% (v/v) 0.1% formic acid in water, 75% (v/v) 0.1% formic acid in acetonitrile.

**Peptide gradient.** The gradient profile was as follows: 0 min 98% A, 2% B; 5 min 98% A, 2% B; 5.5 min 92% A, 8% B; 93 min 60% A, 40% B; 94 min 5% A, 95% B; 104 min 5% A, 95% B; 105 min 98% A, 2% B; 120 min 98% A, 2% B.

**Glycopeptide gradient.** The gradient profile was as follows: 0 min 98% A, 2% B; 6 min 98% A, 2% B; 114 min 60% A, 40% B; 115 min 95% A, 5% B; 119 min 5% A, 95% B; 120 min 98% A, 2% B: 140 min 98% A, 2% B.

**Unmodified peptide identification.** MS1 scans were collected with a mass range from 350 to 1500 $m/z$, 120 K resolution, $4 \times 10^5$ ion inject target, and 50 ms maximum inject time. Dynamic exclusion was set to exclude for 20 s with a repeat count of 1. Charge states 2–6 with an intensity greater than $1\,e^4$ were selected for fragmentation at top speed for 3 s. Selected precursors were fragmented using HCD at 30% nCE 1.2 Da isolation window, $1 \times 10^4$ inject target, and 100 ms maximum inject time. MS2 scans were taken in the ion trap at a rapid scan rate.

**Glycopeptide identification.** MS1 scans were collected with a mass range from 300 to 1500 $m/z$, 120 K resolution, $4 \times 10^5$ ion inject target, and 50 ms maximum inject time. Dynamic exclusion was set to exclude for 10 s with a repeat count of 3. Charge states 2–6 with an intensity $>1 \times 10^4$ were selected for fragmentation at top speed for 3 s. Selected precursors were fragmented using HCD at 28% nCE with 2 Da isolation window, $5 \times 10^4$ inject target, and 54 ms maximum inject time before collection at 30 K resolution in the Orbitrap. For precursors from 300 to 1000 $m/z$, presence of 3 oxonium ions over 5% relative abundance triggered a charge calibrated ETD scan to be collected in the ion trap with a 3 Da isolation window, $1 \times 10^4$ inject target, and 100 ms maximum injection time.

**Unmodified peptide analysis.** Raw MS files were loaded into Max-Quant software (version 1.6.5.0)[55] for quantification and identification by using Homo sapiens (downloaded 18th June, 2020) and Mus musculus (downloaded 10th September, 2020) FASTA protein sequences database from UniProt as a reference database[56]. For peptides, search parameters included specific cleavage specificity of R and K, with two missed cleavages allowed. Methionine oxidation and N-terminal acetylation were set as variable modifications with a total common max of 5. Carbamidomethyl cysteine was set as a fixed modification. Minimum peptide length was set at 7 amino acid units. Mass tolerance was set at 20 ppm for MS1s, 10 ppm for HCD MS2s. Peptide- and protein-level FDR were set at 0.01. The protein groups and peptides tables were uploaded into Perseus (version 1.6.14.0)[57] to allow for data transformation, visualization and statistical analysis. Differentially enriched peptides were determined using Welch's $T$-test (two-sided). Peptide hits were filtered using an eightfold enrichment and $p$ value of 0.1.

**Glycopeptides analysis.** Data evaluation was performed with Byonic™ (Protein Metrics, Cupertino, USA, version 4.0.12). For glycopeptide analysis, search parameters included semi-specific cleavage specificity at the C-terminal site of R and K, with two missed cleavages allowed. Mass tolerance was set at 10 ppm for MS1s, 20 ppm for HCD MS2s, and 0.2 Da for ETD MS2s. Carbamidomethyl cysteine was set as a fixed modification. Variable modifications included methionine oxidation (common 1), asparagine deamidation (common 1), and a custom database of O-glycans that included HexNAc, HexNAc-NeuAc, HexNAc-Hex, HexNAc-Hex-NeuAc, HexNAc2-Hex-NeuAc and HexNAc-Hex-NeuAc2 with an additional 287.1371 $m/z$ to account for the chemical modification. A maximum of two variable modifications were allowed per peptide. For each sample, variable modifications were searched against a focused FASTA file that exclusively contains protein sequences find in that sample. All identifications with |logP| >3 that contained chemically modified glycans were manually validated and localised using a combination of HCD and ETD information.

### Glycome analysis by mass spectrometry

For N -and O-glycan structural analysis, all cells were treated as described previously[59]. Briefly, cell pellets (10,000,000 cells/sample) were subjected to sonication in the presence of CHAPS detergent (Thermo Fisher), reduced in 4 M guanidine-HCl (Pierce), carboxymethylated, and digested with Trypsin gold Mass Spec Grade. The digested glycoproteins were then purified by an Oasis HLB Sep-Pak (Waters). N-glycans were released by peptide N-glycosidase F (E. C. 3.5.1.52; Roche Applied Science, Penzberg, Germany) digestion, whereas O-glycans were released by reductive elimination. Released N- and O-glycans were permethylated using the sodium hydroxide procedure and purified by $C_{18}$-Sep-Pak (Waters).

MS and MS/MS data were acquired using a 4800 MALDI-TOF/TOF (Applied Biosystems) mass spectrometer. Permethylated samples were dissolved in 10 µL of methanol, and 1 µL of dissolved sample was premixed with 1 µL of matrix (10 mg/ml of 3,4-diaminobenzophenone in 75% (v/v) aqueous Acetonitrile), spotted onto a target plate, and dried under vacuum.

The MS and MS/MS data were processed using Data Explorer 4.9 Software (Applied Biosystems). The processed spectra were subjected to manual assignment and annotation with the aid of a glyco-bioinformatics tool, GlycoWorkBench[60]. The proposed assignments for the selected peaks were based on $^{12}$C isotopic composition together with knowledge of the biosynthetic pathways. The proposed structures were then confirmed by data obtained from MS/MS analysis experiments.

### Mouse experiments

NOD-SCID IL2Rgnull (NSG) strain mice (strain nomenclature NOD.Cg-PrkdcSCID Il2rgtm1Wjl/Sz) were obtained from the Jackson Laboratory and bred at the Francis Crick Institute Biological Resources Facility in individually vented cages under specific-pathogen-free conditions at light/dark cycle 7-7, 21 °C and at 50% humidity. When performing the mammary fat pad injection of cancer cells into the mice, the mice were anaesthetised with inhaled isoflurane.

All animals in the experiments discussed here were performed under project license (P83B37B3C), approved by the UK Home Office, and in accordance with The Francis Crick Institute animal ethics committee guidelines. Tumour burden limit for the given experiments is 12 mm × 12 mm. Such maximal tumour size was not exceeded.

### Bioorthogonal cell-specific tagging of glycoproteins in vivo

GFP-expressing 4T1 cells transfected with pSBbi-AGX1$^{F383A}$-NahK-T2$^{BH}$ were resuspended in 50 µL of growth-factor-reduced Matrigel (BD Biosciences) and injected into the left hind fat pad of 9 female NSG mice (6–8 weeks old), with GFP-expressing 4T1 cells transfected with empty pSBbi-Hyg injected into the right hind fat pad of the mouse as an internal control (1,000,000 cells per injection). Body weight and tumour volume were monitored every other day before compound

treatment and did not vary markedly between groups. Mice were randomly assigned to treatment groups based on tumour size.

**Intratumoural compound administration.** 11 days post injection, 3 mice harbouring tumours of roughly 200–300 mm³ were randomly assigned to receive intratumoral injection with 50 μL of vehicle (5% (v/v) DMSO/PEG-400; $n = 1$) or Ac₄GalN6yne (6 mg/mL in 5% (v/v) DMSO/PEG-400; $n = 2$) once daily for 3 consecutive days. On the fourth day the tumours were harvested, flash-frozen and homogenised for Western blot analysis. In the case of the BOCTAG-T2 tumours, an additional piece was prepared for tumour digestion, protein expression analysis and metabolic cell surface labelling (see below for procedures).

**Intraperitoneal compound administration.** 13 days post tumour cell injection, 6 mice were randomly divided into three groups and intra-peritoneally injected with 100 μL of vehicle (5% (v/v) DMSO/PEG-400; $n = 1$), Ac₄GalN6yne (40 mg/mL in 5% (v/v) DMSO/PEG-400; $n = 3$) or Ac₄ManNAlk (40 mg/mL in 5% (v/v) DMSO/PEG-400; $n = 2$) once daily for 5 consecutive days. On the sixth day, the tumours were harvested, flash-frozen and homogenised for Western blot analysis. In the case of the BOCTAG-T2 tumours, an additional piece was prepared for tumour digestion, protein expression analysis and metabolic cell surface labelling (see below for procedures).

**Tumour homogenization and Western blot analysis**
Tumour pieces were homogenised in ice-cold Lysis Buffer (50 mM Tris-HCl pH 8, 150 mM NaCl, 1% (v/v) Triton X-100, 0.5% (v/v) sodium deoxycholate, 0.1% (w/v) SDS, 1 mM MgCl₂, and 100 mU/μL benzonase) containing 50 μM of PUGNAc using a Precellys© homogeniser (speed = 4000 $g$, number of cycles = 3, cycle duration = 30 s, waiting time between 2 cycles = 30 s, temperature = 4 °C). The samples were allowed to stand for 10 min and subsequently centrifuged (10,000 $g$, 4 °C, 15 min) to remove cell debris. The supernatant was collected and the protein concentration determined by BCA assay.

To remove endogenous biotinylated proteins, the samples were diluted up to 250 μL with PBS and incubated 2h at RT with 300 μL of Neutravidin-Coated Magnetic Beads slurry previously washed with PBS (3 × 200 μL). The supernatant was subsequently collected and the protein concentration determined by Pierce™ BCA Protein assay kit.

The collected supernatant (30 μg protein) was then treated with a freshly prepared CuAAC solution containing 600 μM BTTAA, 300 μM CuSO₄, 5 mM sodium ascorbate, 5 mM aminoguanidinium chloride and 100 μM biotin-picolyl azide and incubated overnight at RT on an orbital shaker.

Loading buffer was subsequently added to the samples and these were run on a 4–20% Criterion™ TGX™ Precast gel for SDS-PAGE. After transferring proteins on a nitrocellulose membrane, the total protein amount was assessed using the REVERT protein staining kit and bio-tinylation detected using IRDye 800CW Streptavidin according to the manufacturer's instructions.

**Metabolic labelling of BOCTAG-T2 cells before and after in vivo injection**
The BOCTAG-T2 tumours were manually minced with a scalpel and scissors until they became a smooth paste with no visible clumps and then incubated with 1.5 mL of a freshly prepared digestion solution (a 1:3:3:193 mixture of DNase I, Liberase TM, Liberase TH and HBSS without CaCl₂ or MgCl₂) for 2 h at 37 °C. The cell suspension was then filtered using a 100 μm cell strainer and the reaction quenched by adding an equal volume of growth medium (DMEM with 10% (v/v) FBS, penicillin (100 U/mL), streptomycin (100 μg/mL)) to the filtered cell suspension. The samples were subsequently centrifuged (300 $g$, 10 min, 4 °C) and the pellet resuspended in 10 mL growth medium. The cell suspension was then plated on a 10-cm dish and incubated overnight at 37 °C. After 24 h, the cells were washed once with PBS and treated with fresh growth medium. The cells were subsequently propagated in fresh growth medium with 50 μg/mL hygromycin B at 37 °C with 5% CO₂ for 10 days.

For protein expression analysis and metabolic cell surface labelling, the cells were seeded at a density of 500,000 cells/mL in 1.6 mL growth medium without hygromycin into 6-well plates and allowed to attach overnight. The cells were then treated with 50 μM Ac₄GalN6yne or 50 μM Ac₄ManNAlk or a corresponding volume of DMSO. After 20 h, the cells were washed once with PBS (without CaCl₂ or MgCl₂) and incubated with 8 mM EDTA in PBS (1.5 mL) for 5 min. The cells were transferred to a 1.5 mL centrifuge tube, harvested (500 $g$, 5 min, 4 °C) and resuspended in ice-cold Labelling Buffer (0.2 mL). The metabolic cell surface labelling and in-gel fluorescence analysis was subsequently performed as explained above.

Protein expression was assessed by Western blot using antibodies against FLAG tag (rabbit anti-FLAG antibody, 1:1000, PA1-984B, Invitrogen, Carlsbad, USA), HA tag (rabbit anti-HA antibody, 1:1000, ab9110, Abcam), VSV-G tag (goat anti-VSV-G, 1:2000, ab3861, Abcam) and GADPH (rabbit anti-GAPDH, 1:5000, ab181602, Abcam).

**Reporting summary**
Further information on research design is available in the Nature Research Reporting Summary linked to this article.

## Data availability
The synthetic compound Ac₄GalN6yne is available from the corresponding author upon request of reasonable quantities as long as stocks last. Mass spectrometry proteomics and glycoproteomics data generated in this study have been deposited in the PRIDE database[61] under accession code PXD035430, PXD035437, PXD035438, PXD035445 and PXD035449.

Mass spectrometry Glycomics data generated in this study have been deposited in the GlycoPost[62] database under accession code GPST000293.

RNA-sequencing data generated in this study have been deposited in the GEO database under accession code GSE213052.

The experimental data that support the findings of this study are available from the corresponding author upon request without any reservation. The source data underlying Figures and Supplementary Figures are provided as a Source Data file. Source data are provided with this paper.

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

## Acknowledgements
We thank Kayvon Pedram for providing StcE and Junwon Choi for UDP-sugar standards for chromatography. We thank Lucia Di Vagno for help with glycoproteomics data, Phil Walker for advice on vector choice, and Rocco D'Antuono and Kurt Anderson of the Crick Advanced Light Microscopy STP for support and assistance in this work. We thank Luiz Pedro Carvalho for helpful support, and Jerome Nicod, Robert Goldstone and Phil East of the of the Advanced Sequencing Facility and Bioinformatics and Biostatistics Science Technology Platform for help with transcriptomics samples preparation, sequencing and data analysis. We are grateful for support by the Francis Crick Institute Cell Services and Peptide Chemistry Science Technology Platforms. We thank Julie Bee and Tim Zverev from the Biological Resources Unit at the Francis Crick Institute for technical support with mice and mouse tissue. This work was supported by the Francis Crick Institute (A.C., B.C., T.R., V.B., A.M., G.B.-T., H.F., Z.L., O.Y.T., C.R., P.S.-B., S.K., I.M., B.S.) which receives its core funding from Cancer Research UK (FC001749, FC001060, FC001112), the UK Medical Research Council (FC001749, FC001060, FC001112 and Wellcome Trust (FC001749, FC001060, FC001112). This work was supported by the ERC (788231 to S.L.F.), the Wellcome Trust (218304/Z/19/Z to A.M. and B.S.), the EPSRC (EP/S013741/1 to T.K. and M.A.F., and EP/S005226/1 to S.L.F.), the BBSRC (BB/T01279X/1 to Z.L. and B.S., BB/M027791/1 and BB/M028836/1 to S.L.F., and BB/M02847X/1 to T.K. and M.A.F.) and the NIH (R01 CA200423 to C.R.B.). B.C. was supported by a Crick-HEI studentship funded by the Department of Chemistry at Imperial College London and the Francis Crick Institute. S.A.M. is supported by the Yale Science Development Fund and NIGMS R35 GM147039. K.E.M. is supported by a Yale Endowed Postdoctoral Fellowship and S.C.L. is supported by the National Institutes of Health Chemical Biology Training Grant (T32 GM067543). For the purpose of Open Access, the author has applied a CC BY public copyright licence to any Author Accepted Paper version arising from this submission.

## Author contributions
A.C., B.C., T.R., V.L.B., A.M., G.B.-T., H.F., A.G.-G., A.A. performed experiments; Z.L., O.Y.T., C.R., T.M.W., T.K., P.B., K.H., F.P., S.K., M.A.F., C.R.B., S.L.F. made and provided cell lines, recombinant proteins and compounds; A.C., B.C., S.C.L., K.E.M., A.M., G.B.-T., P.S.-B., M.-R. R., A.A., A.P.S., M.S., S.M.H., S.A.M., I.M., B.S. analysed data; A.C., B.C., K.E.M., S.A.M., I.M. and B.S. wrote the paper with input from all authors.

## Funding

## Competing interests
C.R.B. is a cofounder and scientific advisory board member of Lycia Therapeutics, Palleon Pharmaceuticals, EnableBioscience, Redwood Biosciences (a subsidiary of Catalent), OliLux Biosciences, Grace Science LLC, InterVenn Biosciences, Virsti Therapeutics, and GanNA Bio, and a scientific advisory board member of Rayze Bio, Spotlight Therapeutics, Ambigon Therapeutics, Jupiter Therapeutics, Mekonos, Shasqi Pharma, Ono Pharmaceuticals, Elysium Health, and the Glenn Foundation. S.A.M. is a consultant for InterVenn Biosciences and Arkuda Therapeutics. Other authors declare no competing interests.

## Additional information

[1]Department of Chemistry, Imperial College London, London W12 0BZ, UK. [2]Chemical Glycobiology Laboratory, The Francis Crick Institute, London NW1 1AT, UK. [3]Tumour-Host Interaction Laboratory, The Francis Crick Institute, London NW1 1AT, UK. [4]Department of Chemistry, Yale University, New Haven CT 06511, USA. [5]Proteomics Science Technology Platform, The Francis Crick Institute, London NW1 1AT, UK. [6]Structural Biology Science Technology Platform, The Francis Crick Institute, London NW1 1AT, UK. [7]Bioinformatics & Biostatistics Science Technology Platform, The Francis Crick Institute, London NW1 1AT, UK. [8]RNA Networks Laboratory, The Francis Crick Institute, London NW1 1AT, UK. [9]Mycobacterial Metabolism and Antibiotic Research Laboratory, The Francis Crick Institute, London NW1 1AT, UK. [10]Department of Life Sciences, Imperial College London, London SW7 2AZ, UK. [11]Sarafan ChEM-H, Department of Chemistry and Howard Hughes Medical Institute, Stanford University, Stanford CA 94305, USA. [12]Department of Chemistry, University of York, York YO10 5DD, UK. [13]School of Chemistry & Institute of Biotechnology, The University of Manchester, Manchester M1 7DN, UK. [14]Present address: Massachusetts Institute of Technology, Cambridge MA 02139, USA. [15]Present address: R&D Department, Axxence Slovakia s.r.o., 81107 Bratislava, Slovakia. [16]Present address: Department of Chemistry and Biochemistry, University of Maryland, College Park MD 20742, USA. [17]Present address: Department of Chemistry, Materials and Chemical Engineering "G. Natta", Politecnico di Milano, 20131 Milano, Italy. [18]These authors contributed equally: Tatiana Rizou, Sarah C. Lowery, Victoria L. Bridgeman, Keira E. Mahoney. ✉e-mail: b.schumann@imperial.ac.uk

