## [Peer Review File · Nature Communications]

REVIEWERS' COMMENTS

Reviewer #3 (Remarks to the Author):

My previous critiques of this manuscript focused dominantly on ensuring access to mass spectrometry and glycomics datasets. The authors have described a rigorous approach to providing these data and have completely addressed concerns I had on this front. They adequately addressed my other comments, as well. I enthusiastically endorse publication of the revised manuscript in its current form.

At the request of the editor, I also considered the response of the authors to Referee 1's critiques. The authors systematically addressed this referee's critiques by performing a battery of experiments (e.g., measuring nucleotide-sugar levels and probing the potential for altered flux/feedback inhibition, performing a suite of lectin and immunoblots to acquire complementary data to probe glycans, adding information about mouse health/behavior during in vivo labeling experiments) and making appropriate clarifying changes to the text (e.g., indicating that multiple classes of glycans are metabolically labeled using this approach). They also, quite reasonably, point out that many of the limitations flagged by Referee 1 (e.g., artificial enhancement of levels) occur in well-established techniques in cancer biology, including in overexpression and knockout experiments. Thus, in addition to quantifying the very modest amount of flux introduced in glycan expression levels in this system, the authors indicate that alterations are an established part of very powerful genetics toolkits in cancer biology. We can learn important lessons about the underlying (glyco)biology in spite of minor changes in expression levels.

As the result of the peer review process, this manuscript has been considerably strengthened. I enthusiastically recommend publication of the revised manuscript in *Nature Communications*, which I believe is an appropriate venue given the impact of this work. This methodology is a first-in-class approach to query cellular glycoproteomes in a cell-selective manner in complex environments, including within animals. The level of control and precision introduced here to label, track, and identify glycoproteins within select cells in cancer models is unprecedented and will allow the field to probe long-observed but not fully understood changes in glycoprotein expression that track with the cancer phenotype.